# Learning to grok: Emergence of in-context learning and skill composition in modular arithmetic tasks

**Tianyu He** [a, †]        **Darshil Doshi** [a]        **Aritra Das** [a]        **Andrey Gromov** [b, a]

{tianyuh, ddoshi, aritrad}@umd.edu        gromovand@meta.com

## Abstract

Large language models can solve tasks that were not present in the training set. This capability is believed to be due to in-context learning and skill composition. In this work, we study the emergence of in-context learning and skill composition in a collection of modular arithmetic tasks. Specifically, we consider a finite collection of linear modular functions $z = a\,x + b\,y \bmod p$ labeled by the vector $(a, b) \in \mathbb{Z}_p^2$. We use some of these tasks for pre-training and the rest for out-of-distribution testing. We empirically show that a GPT-style transformer exhibits a transition from in-distribution to out-of-distribution generalization as the number of pre-training tasks increases. We find that the smallest model capable of out-of-distribution generalization requires two transformer blocks, while for deeper models, the out-of-distribution generalization phase is *transient*, necessitating early stopping. Finally, we perform an interpretability study of the pre-trained models, revealing highly structured representations in both attention heads and MLPs; and discuss the learned algorithms. Notably, we find an algorithmic shift in deeper models, as we go from few to many in-context examples.

## 1   Introduction

Large language models (LLMs) can perform simple tasks absent from their training data. This ability is usually achieved via in-context learning [5]. More importantly, LLMs can perform an even larger variety of very complex tasks upon appropriate prompting or fine-tuning. The latter ability to perform complex tasks is usually attributed to the following mechanism. First, LLMs learn a large variety of simple tasks and, then, how to *compose* those skills to form very complex skills [3]. Furthermore, LLMs also exhibit "emergent capabilities" – a sudden emergence of a new complex skill as a function of scale (either model size, compute or data) [29, 9]. It is plausible that these sudden performance improvements are due to one or both of these mechanisms. For example, LLMs show grokking on algorithmic tasks [24], which results from the model learning very structured representations [18, 11, 20]. Once these representations emerge, the model abruptly learns how to perform the task.

In this work, we set out to examine skill composition both empirically and mechanistically. Inspired by the prior work that investigated emergence of in-context learning on linear regression tasks [1, 25], we introduce a finite collection of discrete modular arithmetic tasks [24] generalized to the in-context learning setting. Each task corresponds to learning a linear function $z = a\,x + b\,y \bmod p$ over $\mathbb{Z}_p$ from the examples provided in context of the autoregressive model(AM). In the bi-variate case there are $p^2$ such functions labeled by the vector $(a, b)$. The main objective of this algorithmic dataset is to probe *how* AM utilizes the tasks it has learnt during training to solve the new tasks.

---

[a]Department of Physics, University of Maryland, College Park

[b]Meta AI

[†]Corresponding author

[*]Source Code: https://github.com/ablghtianyi/ICL_Modular_Arithmetic

38th Conference on Neural Information Processing Systems (NeurIPS 2024).

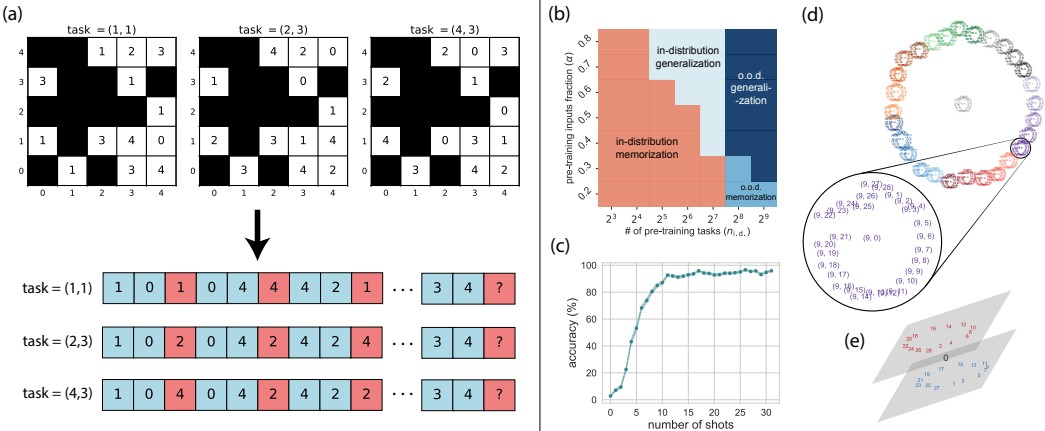

Figure 1: **(a)** The dataset. The tasks are labeled by vectors $(a, b) \in \mathbb{Z}_p^2$. Each table contains examples of $ax + by$ mod $p$. A fraction $1 - \alpha$ of the examples is blacked out; while the remaining examples are flattened into a single "document" in the batch. Each document is organized as a collection of triples $(x, y, ax + by)$ for $x, y$ from the training set (*i.e.* not blacked out in the table). Our training is similar to the traditional next-token prediction (autoregressive); with the main difference that we predict every third token, which are marked in red ($x$ and $y$ are uncorrelated). Every task appears exactly the same number of times in each batch. **(b)** Phase diagram for a six-layer model. We find four different phases. (1) *in-distribution memorization*: The model *only* performs well on tasks $(a, b)$ *and* examples $(x, y)$ from the training set – it does not generalize on unseen examples or tasks. (2) in-distribution generalization: model generalizes on unseen examples $(x, y)$ but not on unseen tasks $(a, b)$. (3) out-of-distribution memorization: model generalizes on unseen tasks $(a, b)$ but only for examples $(x, y)$ it has seen during training. (4) out-of-distribution generalization: model generalizes on unseen tasks $(a, b)$ for seen as well as unseen examples $(x, y)$. We focus on investigating phase (4) in more detail. **(c)** In-context sample complexity. Accuracy of the model in phase (4) as a function of the number of few-shot examples. **(d)** Representations developed by one of the attention heads in the first layer. These are projections of the embedding of a pair of numbers onto the two largest principal components (PCs) of the internal representation formed after passing through the attention layer and projection matrix. **(e)** First 3 PCs of embeddings separate $log_{27}$-annotated numbers into even/odd planes, with 0 sandwiched between them.

Our analysis shows that the solution found by the AM after optimization is qualitatively different from the linear regression cases studied before [1]. In those cases, due to the continuous nature of the task, AM develops an emergent first-order optimization method that minimizes an emergent quadratic loss function. Furthermore, as it was shown in [1], a single linear attention layer can solve the regression problem, while adding extra layers and non-linearities slightly modifies the gradient descent. In the modular arithmetic case, AM first learns how to solve the pre-training tasks and later (assuming enough different tasks) develops a generalizing solution by combining the solved tasks.

Our main findings as well as the structure of the algorithmic dataset are illustrated on Fig. 1. Our main findings are: (i) there are **four** different phases in the end of pre-training depending on the number of tasks, $n_{\text{i.d.}}$, and number of examples per task, $\alpha$. (ii) At inference time, there is a generalization transition in the number of few-shot examples, as the number of examples grows, the models starts to generalize. This effect is somewhat similar to the transition in sample complexity for the modular arithmetic found in [24]. (iii) model develops a striking circular representation for all of the tasks that naturally generalizes the circular representations found in the original work [24]. We further find that the deeper models are easier to optimize, but much harder to interpret. The optimization is discussed in more detail in the main text. Here we will highlight that optimization for these tasks is challenging and the AM tends to prefer the minimum that just solve a handful of tasks and memorize the training set. To avoid such minima we make sure that *every batch* contains equal number of tasks (meaning that no tasks is over- or under-represented in each batch). We further find that for larger models early stopping is necessary because the generalizing solution is transient.

We organize our paper as follows. Section 2 contains the literature review. In Section 3 we explain our notations and discuss the experimental details. In Section 4 we demonstrate empirically that the

out-of-distribution ICL ability emerges as the number of training tasks increases. We also study the effects of model depth and task difficulty. In Section 5 we carefully examine a minimal setting, *i.e.* two-block transformer: we compare the representations learnt in four different phases and show that in the generalizing phase the representations are highly structured and generalize the original modular addition case of [24].

## 2   Related Works

**In-Context Learning (ICL)**   Brown et al. [5] first demonstrated that large models performance improves substantially when a few examples of the task at hand are provided at inference time, in the prompt. Akyürek et al. [2], Ahn et al. [1], von Oswald et al. [28] showed that the AM implements emergent first-order optimization on an emergent objective function to solve linear regression tasks. Furthermore, [2] showed that larger models learn to perform Bayesian estimation. Garg et al. [10] demonstrated that transformers can learn several simple classes of functions in context. Kirsch et al. [15] presented how task diversity and model size would affect the ICL performance for unseen tasks using a mixture of modified MNIST datasets. Raventos et al. [25] investigated the relation between task diversity and out-of-distribution ICL ability on linear regression tasks. Lin and Lee [16] identified two operating modes in ICL using a mixture of linear regression tasks, where for the first several shots, the model tries to figure out the correct task vector and later uses it to predict the correct results. Boix-Adserà et al. [4] showed theoretically and experimentally that with enough pre-training data, a transformer model can perform abstract reasoning that a MLP cannot do. Guo et al. [13] showed that transformers can use lower layers to memorize and upper layers to perform ICL in a feature regression setting. It was found in [26] that ICL is a transient phase from the optimization point of view: it goes away once the model is over-trained. Hendel et al. [14], Liu et al. [17] showed that language models form in-context vectors, which can be used to steer model predictions.

**Modular Arithmetic**   Power et al. [24] discovered Grokking, where models trained on modular arithmetic datasets have an abrupt change from random guessing to generalization on the test set way after the model memorized the training set. Gromov [11], Nanda et al. [20], Gu et al. [12] showed that for modular addition tasks, models learned to map integers to Fourier features to solve modular arithmetic tasks. Liu et al. [18] showed that grokking is related to learning highly structural features, and the grokking transition can be explained by a toy model. Zhong et al. [30] showed that there is more than one algorithm that a model can implement to solve modular addition. Doshi et al. [6] showed that corruption of the label does not prevent the models from finding a generalizing solution. Doshi et al. [7] showed that MLP and transformer models can solve a specific family of modular polynomial tasks by bijectively mapping them to modular addition tasks.

**Interpretability**   Elhage et al. [8], Olsson et al. [22] showed that transformers can form induction heads that predict the next token in a sequence by identifying and copying patterns from earlier in the sequence. With several indirect empirical evidences, they showed that those heads might constitute the core mechanism of ICL. Nichani et al. [21] showed theoretically and empirically how disentangled transformers learn causal structures from in-context Markov chains by forming induction heads.

## 3   Preliminaries

**Linear Modular Functions**   We consider modular arithmetic tasks of the form: $z_i^t = a^t x_i + b^t y_i \mod p$. We will refer to the coefficients $(a^t, b^t) \in \mathbb{Z}_p^2$ as the *task vector*. The superscript $t \in \{1, \cdots, p^2\}$ labels the $p^2$ possible tasks. We will refer to $(x_i, y_i) \in \mathbb{Z}_p^2$ as the *input vector*, which is labeled by the subscript $i \in \{1, \cdots, p^2\}$.

**In-Context Learning with Transformers**   We use GPT-like transformers [5] with ReLU activation function and Rotary Positional Embedding (RoPE) [27]. The model has $d$ consecutive blocks, $H$ attention-heads, and embedding dimension $d_{\text{embed}}$. Each number is tokenized as an independent token. The pre-training is done following a slightly modified next-token prediction setup, with sequences of the form:

$$\boldsymbol{s}^t = \begin{pmatrix} x_1 & y_1 & z_1^t & x_2 & y_2 & z_2^t & \cdots & x_{n_{\text{ctx}}} & y_{n_{\text{ctx}}} & z_{n_{\text{ctx}}}^t \end{pmatrix} \in \mathbb{Z}_p^{3 \times n_{\text{ctx}}}, \tag{1}$$

where $n_{\text{ctx}}$ is the maximum number of in-context examples. The model is asked to predict *only* the labels $\{z_1^t, \cdots, z_{n_{\text{ctx}}}^t\}$. We emphasize that we do not explicitly provide the task vectors $(a^t, b^t)$ to the model (see Fig. 1) – this information is implicit in the in-context labels $\{z_i^t\}$. In order for the model to generalize, it must determine the underlying task vector $(a^t, b^t)$ from the few-shot examples.

**Generalization** There are two notions of generalization in this setup. (i) *In-distribution*: Generalization to *unseen* input vectors $(x_i, y_i)$, but on task vector $(a^t, b^t)$ the model has seen during pre-training. (ii) *Out-of-distribution*: Generalization to task vectors the model has *not* seen during pre-training. To clearly separate these regimes, we split the task vectors into in-distribution (i.d.) set $\mathcal{T}_{\text{i.d.}} := \{(a^t, b^t)\}_{\text{i.d.}}$ and out-of-distribution (o.o.d.) set $\mathcal{T}_{\text{o.o.d.}} := \{(a^t, b^t)\}_{\text{o.o.d.}}$. Similarly, we split the input vectors into train and test

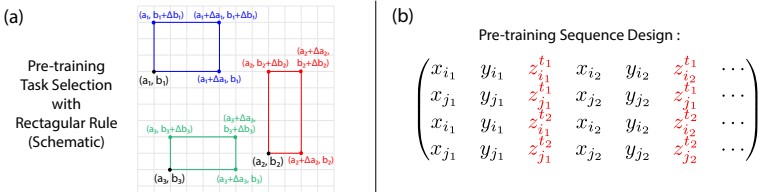

| | | $S_{train}^{i.d.}$ | $S_{test}^{i.d.}$ | $S_{train}^{o.o.d.}$ | $S_{test}^{o.o.d.}$ |
|---|---|---|---|---|---|
| In-distribution memorization | | ✅ | ❌ | ❌ | ❌ |
| In-distribution generalization | | ✅ | ✅ | ❌ | ❌ |
| Out-of-distribution memorization | | ✅ | ❌ | ✅ | ❌ |
| Out-of-distribution generalization | | ✅ | ✅ | ✅ | ✅ |

✅ = performs well    ❌ = performs poorly

sets: $\mathcal{X}_{\text{train}} := \{(x_i, y_i)\}_{\text{train}}, \mathcal{X}_{\text{test}} := \{(x_i, y_i)\}_{\text{test}}$. This results in four distinct sets of sequences constructed from those sets; we name them $S_{\text{train}}^{\text{i.d.}}, S_{\text{test}}^{\text{i.d.}}, S_{\text{train}}^{\text{o.o.d.}}$ and $S_{\text{test}}^{\text{o.o.d.}}$. The set $S_{\text{train}}^{\text{i.d.}}$ is used for pre-training, while the other three sets are used for evaluations.

**Pre-Training Task Selection and Sequence Design** We always sample the pre-training task vectors $\mathcal{T}_{\text{i.d.}}$ in sets of 4, following the rectangular rule, shown in Figure 2(a). Additionally, each batch contains an equal representation from all the task vectors in the set $\mathcal{T}_{\text{i.d.}}$. Moreover, all the tasks share the same sequence of inputs. For example, a batch with two different task vectors $(a^{t_1}, b^{t_1}); (a^{t_2}, b^{t_2})$ and two distinct input sequences per task (resulting in four total sequences) is shown in Figure 2(b).

(a)

Pre-training Task Selection with Rectangular Rule (Schematic)

(b)

Pre-training Sequence Design :

$$\begin{pmatrix} x_{i_1} & y_{i_1} & z_{i_1}^{t_1} & x_{i_2} & y_{i_2} & z_{i_2}^{t_1} & \cdots \\ x_{j_1} & y_{j_1} & z_{j_1}^{t_1} & x_{j_2} & y_{j_2} & z_{j_1}^{t_1} & \cdots \\ x_{i_1} & y_{i_1} & z_{i_1}^{t_2} & x_{i_2} & y_{i_2} & z_{i_2}^{t_2} & \cdots \\ x_{j_1} & y_{j_1} & z_{j_1}^{t_2} & x_{j_2} & y_{j_2} & z_{j_2}^{t_2} & \cdots \end{pmatrix}$$

Figure 2: Structured selection of pre-training tasks and sequences.

This structured approach creates a coherent signal from the sequences within each batch; ensuring that the model learns multiple task vectors with reasonable batch sizes. Alternatively, if the batches are sampled i.i.d., then the model is confused by the batch noise and cannot learn any tasks.

Detailed discussions on task selection and sequence design are presented in Appendix D.

**Default Setting** Unless stated explicitly, we will use $p = 29$, the number of heads $H = 4$, and embedding dimension $d_{\text{embed}} = 512$, with $n_{\text{ctx}} = 32$ in-context examples. All models are trained with AdamW optimizer [19] with batch size $B = 1024$ for 200k steps. We have also tied the embedding layer of the model with the readout layer.

## 4 Emergence of In-Context Learning and Task Composition

In this section, we demonstrate that a transformer model with depth $d \geq 2$ can develop ICL and out-of-distribution generalization on modular arithmetic tasks. We delve deeper into the two notions of generalization (i.d. and o.o.d.), and discuss the relevant factors. We find that the model's ability to generalize out-of-distribution is predominantly determined by the number of pretraining tasks $n_{\text{i.d.}}$.

### 4.1 Transition driven by the number of tasks

In Figure 3(a), we show the accuracy of $d = 6$ models vs the number of training tasks $n_{\text{i.d.}}$ and the number of few-shot examples quantified by the fraction of the total number of data points, $\alpha$; on sets $S_{\text{train}}^{\text{i.d.}}, S_{\text{test}}^{\text{i.d.}}, S_{\text{train}}^{\text{o.o.d.}}$ and $S_{\text{test}}^{\text{o.o.d.}}$. The phase diagram in Figure 1 is constructed by merging the last shot accuracy version of these four diagrams shown in Figure 27(a) of Appendix G.

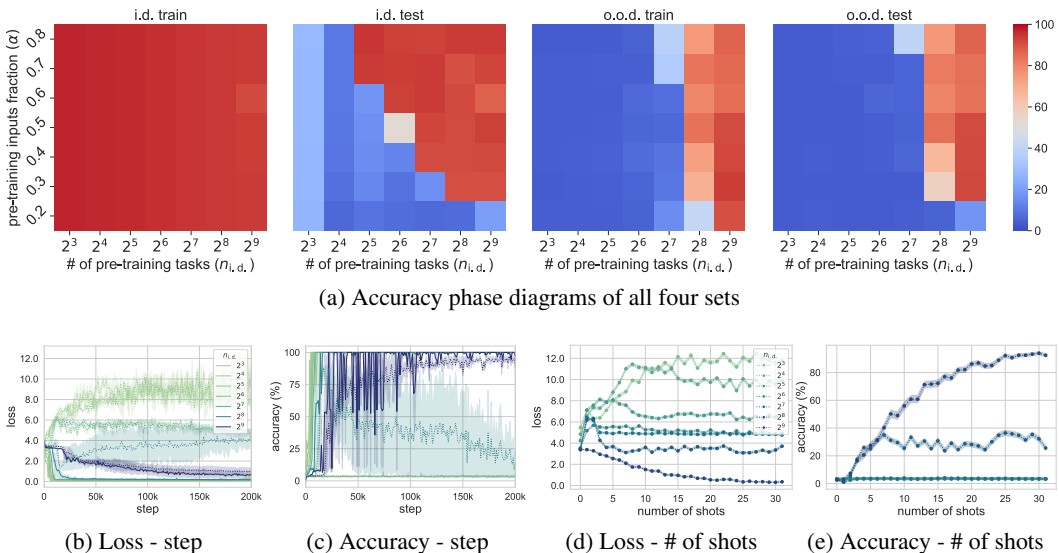

(a) Accuracy phase diagrams of all four sets

(b) Loss - step     (c) Accuracy - step     (d) Loss - # of shots     (e) Accuracy - # of shots

Figure 3: Phase diagram for the depth $d = 6$ models. **(a)** Accuracy on all four sets used to plot the 1 phase diagram, with an early stopping applied. Notably, in the regions when models generalize to o.o.d. sets, the pre-training performance degrades; **(b, c)** $\alpha = 0.6$ training accuracy and o.o.d. test accuracy (dotted line). For $n_{\text{i.d.}} = 2^8$, we notice that the o.o.d. generalization ability of the model first improves then degrades as we train longer; **(d, e)** $\alpha = 0.6$, loss and accuracy vs context length, measured on $S_{\text{test}}^{\text{o.o.d.}}$ at the end of training, where for $n_{\text{i.d.}} = 2^8$ case the ICL ability fades away.

The ability of the model to generalize in-distribution increases with $\alpha$, as can be seen by comparing the first two panels of Figure 3(a). This behavior is in correspondence with the original work on grokking, where the transition to generalizing solution is driven by the amount of data. Further, we observe that an increase in $n_{\text{i.d.}}$ enhances the in-context sample efficiency, *i.e.* the model generalizes at inference time with fewer few-shot examples. This indicates the onset of the transition from a task-memorizing solution to the one that generalizes out-of-distribution. The model switches to a new algorithmic way of solving the task and the solution is more few-shot-sample-efficient.

Shifting our focus to the last two panels of Figure 3(a), we see that when $n_{\text{i.d.}} \geq 256$, the model can solve new tasks that were absent in the training set. Notably, there appears to be a trade-off between memorization and generalization when the model attains this o.o.d. generalization ability. As the o.o.d. performance increases, the pre-training performance simultaneously degrades. This phenomenon indicates a shift in the algorithm implemented by the model. Prior to this transition, the model primarily needed to select possible vectors from the list of memorized tasks and apply them. However, post-transition, the model adopts a more universal approach to solve the task in-context. We emphasize, that the model learns to perform ICL in *both scenarios*. The difference lies in the approach to generalization. When the model can only generalize in-distribution it's task is to classify the sequence as one of the seen tasks or as unknown. Once it matches the sequence to one of the memorized task vectors, it does well for $x, y$ pairs that only appear in the test set. However, as the number of tasks vectors grows the model fails to store them all and is forced to find a method of determining the task vector algorithmically at inference time. In that case model performs equally well on seen and un-seen tasks alike. In fact, the small two-layer model we study has such a low capacity that it entirely skips the in-distribution generalization phase and immediately jumps from pure memorization to out-of-distribution generalization.

Next, to further illustrate the effect of task diversity, we plot the pre-training accuracy (set $S_{\text{train}}^{\text{i.d.}}$) and the o.o.d. test accuracy (set $S_{\text{test}}^{\text{o.o.d.}}$) as a function of training steps (Figure 3(b, c)); for various values of $n_{\text{i.d.}}$. We observe a clear memorization-to-generalization transition as task diversity increases. Interestingly, for $n_{\text{i.d.}} = 2^8$, the ICL ability on $S_{\text{test}}^{\text{o.o.d.}}$ set exhibits non-monotonic behavior, where the o.o.d. performance rises and falls along the training process. This phenomenon is likely due to a competition between the memorizing and generalizing circuits inside the model. Note that this phenomenon is akin to the one analyzed in Singh et al. [26], albeit with a different setup.

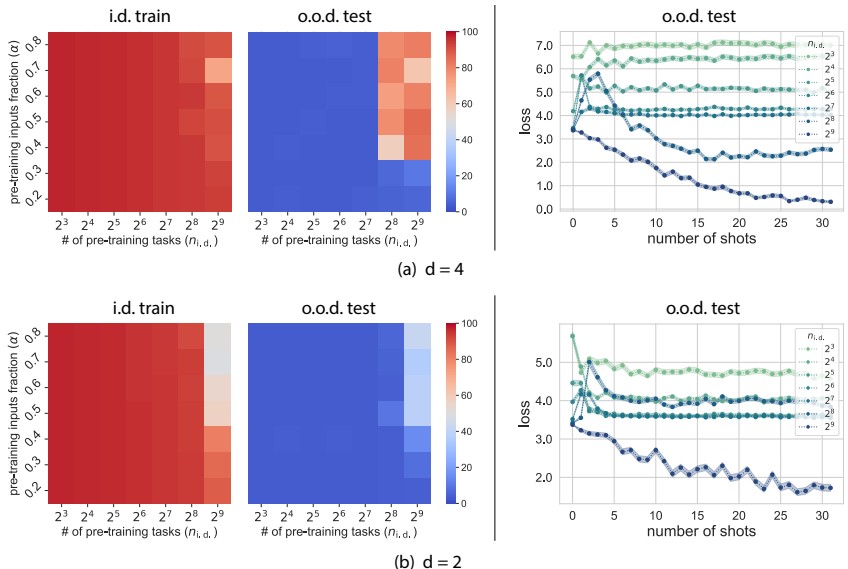

Figure 4: Phases of depth $d = 4$ and $d = 2$ models. With decreasing model capacity, the performance on both sets degrades. At the same time, the transient nature of ICL does not appear. **(a, b) from left to right:** accuracy phase diagrams on pre-training set $S_{\text{train}}^{\text{i.d.}}$ and on o.o.d. test set $S_{\text{test}}^{\text{o.o.d.}}$, with early stopping; loss and accuracy vs context length on o.o.d. test set $S_{\text{test}}^{\text{o.o.d.}}$ for $\alpha = 0.6$.

Further evidence supporting the two-circuit competition can be observed in panel (d) of Figure 3. The loss curves show a "monotonic $\rightarrow$ non-monotonic $\rightarrow$ monotonic" transition as the task diversity increases. With a minimal number of pre-training tasks, the model primarily engages in memorization, resulting in a monotonically increasing o.o.d. loss curve. As the number of pre-training tasks increases, the loss curve exhibits non-monotonic behavior, indicating competition between two distinct neural circuits. This transient nature of o.o.d. generalization for $n_{\text{i.d.}} = 2^8$ is a peculiar case where memorization circuits are initially suppressed but eventually prevail. With substantial task diversity, the circuits responsible for generalization take over, culminating in a monotonic loss curve. Similar insights can be derived from examining the monotonicity of the accuracy curves in panel (e).

## 4.2 Effect of Model Size and Task Difficulty

A natural question to ask is if similar phenomena can be observed with different model sizes or task difficulties. Here we present our results with $d = 4$ and $d = 2$ in Figure 4 and leave the results for other prime $p$ values in Appendix H.

When comparing phase diagrams in Figure 3 with Figure 4, we observe that those phase diagrams across different depths are qualitatively similar, where the o.o.d. generalization only emerges with a large enough number of pre-training tasks. As model capacity decreases, performance on both the pre-training set and the o.o.d. test set degrades. This is particularly evident in the $d = 2$ case, where the pre-training accuracy falls drastically as the model gains o.o.d. generalization ability.

Interesting observations can be made by comparing loss and accuracy on the o.o.d. test set as a function of context length at the end of training. First, it is evident that as the model depth decreases, the 1-shot loss surge attributable to memorization becomes milder. Notably, for models with $n_{\text{i.d.}} = 2^9$, there is no loss surge in the 1-shot case across all three depths. Furthermore, the $d = 4$ model with $n_{\text{i.d.}} = 2^8$ behaves significantly differently from the corresponding one with $d = 2$ case, where the model fails to perform ICL for the o.o.d. test set. This is also distinct from the $d = 6$ case, where the model tends heavily toward memorization due to its excessive capacity. Instead, the $d = 4$ model manages to maintain a better balance between memorization and generalization at the end of pre-training. Consequently, the model has a 1-shot loss surge followed by a notable drop in ICL loss. This suggests that $d = 4$ optimally leverages the available model capacity to facilitate effective learning dynamics for o.o.d. generalization.

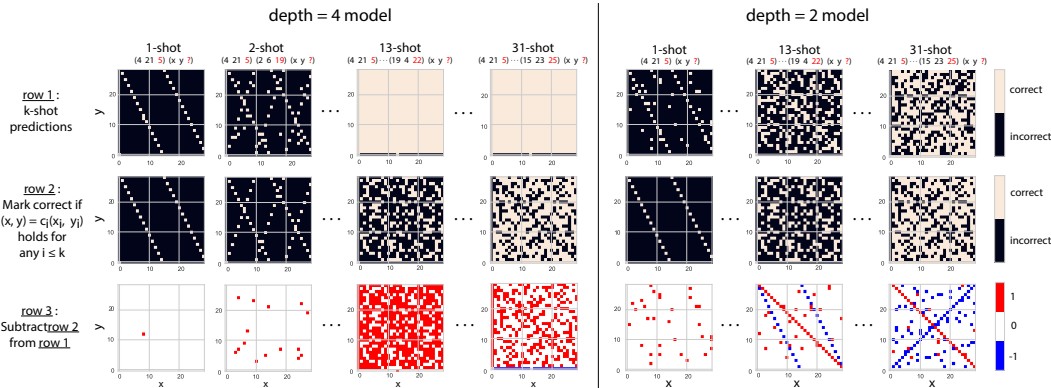

Figure 5: $d = 4$ and $d = 2$ models' performance on k-shot inference, on the grid of inputs $(x, y) \in \mathbb{Z}_p^2$ (task vector $= (6, 6)$). **row 1:** Models' predictions on o.o.d. task of the type $(x_1 \ \ y_1 \ \ z_1 \ \cdots \ x_k \ \ y_k \ \ z_k \ \ x \ \ y \ \ ?)$. **row 2:** Analytical plots showing predictions solely based on Modular Regression algorithm. **row 3:** Subtract row 2 from row 1, by using correct=1 and incorrect=0. The red points correspond to the examples where Ratio Matching does not give the correct predictions but the model predicts correctly. The blue points are examples that the model missed despite Ratio Matching being applicable. This row tells us about the model's ability to implement Modular Regression by *combining* the in-context examples. Note that $d = 4$ model readily learns to combine previous examples, while its $d = 2$ counterpart struggles due to its limited capacity.

## 5 Interpreting the Learned Algorithms

We now explain the algorithms implemented by the models to achieve o.o.d generalization. We find that the models implement two distinct algorithms depending on the depth and the number of in-context examples during inference.

**Ratio Matching:** If there exists $i$ s.t. $c_i(x_i, y_i) = (x, y) \mod p$ then $z = c_i z_i^t \mod p$. This algorithm only works when $y/x = y_i/x_i \mod p$ holds for at least one of the in-context examples.

**Modular Regression:** Find $\{c_1, c_2, \ldots, c_k\}$ s.t. $\sum_{i=1}^{k} c_i(x_i, y_i) = (x, y) \mod p$. Then the prediction is $z = \sum_{i=1}^{k} c_i z_i \mod p$. This can be viewed as discretized circular regression over $\mathrm{GF}(p)$.

We find telling signatures of these algorithms upon analyzing model predictions with varying numbers of in-context examples (Figure 5). With very few in-context examples, the models implement the Ratio Matching algorithm. As a canonical example, consider the 1-shot scenario $(x_1 \ \ y_1 \ \ z_1 \ \ x \ \ y \ \ ?)$. In this case, Ratio Matching will only solve the task for inputs that obey $(x, y) = c_1(x_1, y_1) \mod p$ for some $c_1$. Indeed, this is exactly what we observe in Figure 5 for both $d = 2, 4$ models. With many ($\sim 10$) in-context examples, $d = 2$ and $d = 4$ models implement different algorithms. The $d = 4$ model can *combine* the in-context examples using the Modular Regression algorithm, leading to near-perfect o.o.d. generalization. On the other hand, the $d = 2$ model still uses Ratio Matching and shows sub-optimal performance. We ascribe this to the limited capacity of the $d = 2$ models. In other words, the $d = 4$ models exhibits an *algorithmic shift* from Ratio Matching to Modular Regression as them number of in-context examples increases. Whereas, the $d = 2$ models shows no such shift.

To implement these algorithms, the model needs to perform linear operations over $\mathrm{GF}(p)$, which can be broken down into the following essential skills (in addition to copying information, which is readily implemented by attention heads).

   I. *Modular Map*: Encode the tokens such that operations over $\mathrm{GF}(p)$ can be easily implemented

  II. *Multiplication*: A necessary skill to rescale in-context examples in both algorithms

 III. *Addition*: A necessary skill for combining in-context examples in Modular Regression algorithm

While both $d = 2, 4$ models learn skills I, II perfectly; the $d = 4$ model outperforms its $d = 2$ counterpart in skill III – which helps it *combining* the in-context examples.

In the remainder of this section, we show the special attention heads that implement skill I (Section 5.1) and MLPs that implement skills II, III (Section 5.2). We explicitly show the deterioration in this

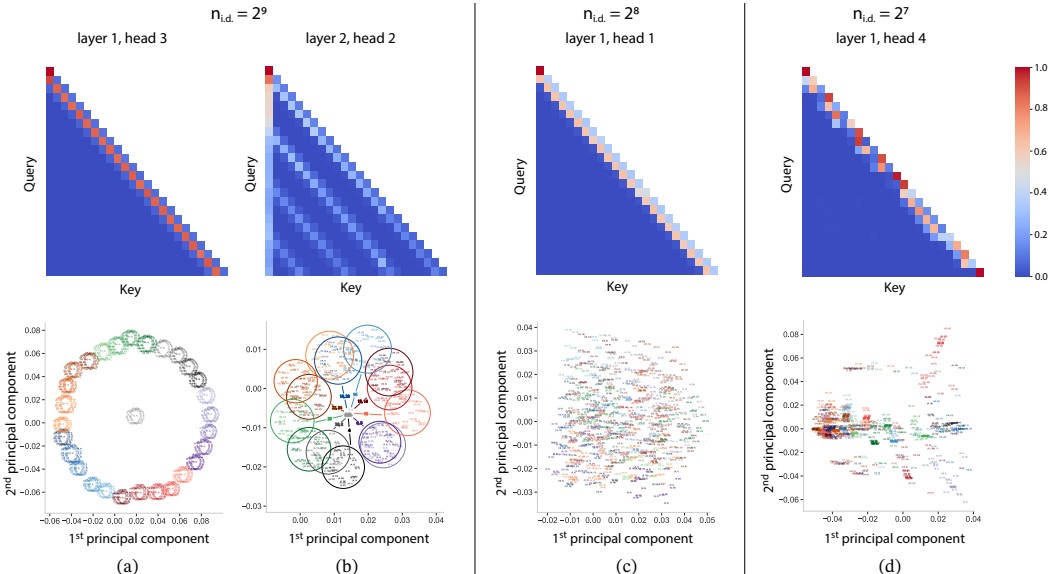

Figure 6: Models that generalize o.o.d. (left) exhibit more structured attention maps. Additionally, the top-2 principal components of the features from the corresponding heads also show more structured patterns. The features are computed for sequences with an o.o.d. task vector $(a^t, b^t) = (6, 6)$, loop over $(x_i, y_i)$ at a specific shot while the previous parts of the sequence are fixed. We annotate each PCA plot with the $(\log_{27} x_i, \log_{27} y_i)$ mod $p$ pairs. **(a)** The principal components form a circle of circles where the position of the outer circle is controlled by $x_i$. This pattern remains the same for different task vectors or the shot choices; **(b)** Only plotted pairs with even $\log_{27} x_i$, with each $\log_{27} x_i$ circled with different colors. The PCA pattern forms a similar double circle as those in (a), with the key difference that those circles depend on task vector choices and the shot choices; **(c, d)** Models without o.o.d. generalization ability. We pick heads from the first block that corresponds to the first column of (a). Clearly, the structure of attention maps and PCA patterns deteriorate as the task diversity decreases.

structures as pre-training task diversity ($n_{i.d.}$) decreases. We also elucidate the algorithmic shift in the $d = 4$ models as the number of in-context examples increases.

## 5.1 Attention Heads Implement Skill I

So far we have presented "black-box" evidence suggesting that the model is implementing the proposed algorithms (Figure 5). Now, we turn to "white-box" interpretability, to identify the components within the transformer model that are responsible for the essential skills. In Figure 6(a, b), we analyze the important attention heads in the $d = 2$ model. Specifically, we compare the emergent structures in models trained with different pre-training task diversities.

In Figure 6(a), we show the attention head from layer 1 that implements skill I. In the top panel, we see that each query only pays attention to itself and the two preceding keys. This pattern likely stems from the fact that each example in the sequence contains three tokens $x_i, y_i, z_i^t$; and suggests that this head is mostly focused on the information within the example.

In the bottom panel, we perform principal component analysis (PCA) on the outputs of this head. Specifically, we feed a batched k-shot sequence of the form $(x_1 \ y_1 \ z_1 \ \cdots \ x_k \ y_k \ z_k \ x \ y \ z)$, where the first $k$ inputs are fixed and the last input $(x, y)$ is scanned over all $p^2$ possible pairs. We concatenate the resulting features from $x$ and $y$, resulting in $p^2 \times 2d_{\text{embed}}$ batch of features – and perform PCA on this matrix. We project all these $2d_{\text{embed}}$ dimensional features onto the first two principal components. Annotating the pairs $(x, y)$ with $(\log_{27} x, \log_{27} y)$[1], we find a "circle-of-circles" – where circles of period 28 are arranged in a bigger circle of period 28. Number 0 is located

---

[1] We use 27 as the base of logarithm, which is a primitive root of $\text{GF}(29)$

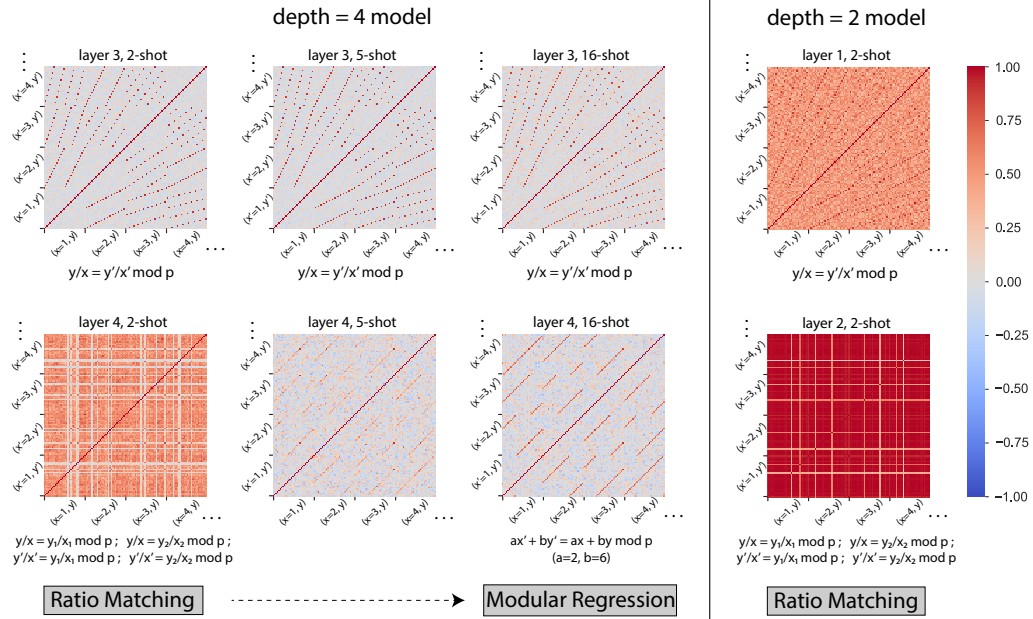

Figure 7: Cosine-similarities (Equation (2)) between layer outputs $\boldsymbol{h}^l$ at token $z$ position $(\cos(\boldsymbol{h}_z^l(x,y), \boldsymbol{h}_z^l(x',y'))$, first row) and token $y$ position $(\cos(\boldsymbol{h}_y^l(x,y), \boldsymbol{h}_y^l(x',y'))$, second row) reveal patterns in the models' internal representations. For clarity, we only show selected $x$ and $x'$ values, where $y$ and $y'$ values range from 0 to 28 between each tick. For the $d = 4$ model, kaleidoscopic patterns in the third layer indicate the generation of all possible $y_i/x_i$ features for subsequent computations. The last layer shows an algorithmic shift from Ratio Matching to Modular Regression. The $d = 2$ model also shows the kaleidoscopic pattern in the first layer, while the second layer identifies the relevant $y/x$ features from the in-context examples for matching.

at the center of the circles[2]. We observe similar circle-of-circles for concatenated features from $(x, z)$ and $(y, z)$ as well. We refer the reader to Appendix E for further details.

In Figure 6(b), we analyze the head from layer 2 that effectively "copies" in-context examples. The upper panel shows the highly structured attention map that focuses on the current as well as the preceding examples. This pattern aligns with the Ratio Matching algorithm where the model compares $(x, y)$ pairs across different in-context examples.

In the bottom panel, by conducting a PCA analysis, we again identify circles when annotating examples in the $(\log_{27} x, \log_{27} y)$ format. However, unlike the previously discussed pattern, the specifics of this "circle-of-circles" arrangement vary depending on the position and the choice of task vector $(a^t, b^t)$. This variability suggests that the head in question possesses information about the specific task from the context.

We further highlight the importance of the structure we find in the above paragraphs via comparison with models, trained with lower pre-training task diversity, that do *not* generalize o.o.d (Figure 6 c, d). Note that as the number of pre-training tasks $n_{\text{i.d.}}$ decreases, the attention map starts to get mosaicked (top panels); and the PCA projections lose their shape (bottom panels). As a result, these models lose the ability to perform ICL on modular arithmetic out-of-distribution.

## 5.2 MLPs Implement Skill II and III

After applying Skill I, the model maps numbers onto circular feature spaces, enabling discrete modular operations such as multiplication, division, addition, and subtraction through subsequent model components. To analyze this behavior, we compute the cosine-similarity between layer $l$ (i.e. $l^{th}$ Transformer block) $k$-shot output vectors (standardized) at token position $y$: $\boldsymbol{h}_y^l(x_1, y_1, z_1, \cdots, x_k, y_k, z_k, x, y)$.

---

[2]This is expected since $\log_{27} 0$ is mathematically ill-defined and needs special treatment.

$$\cos(\boldsymbol{h}_y^l(x,y), \boldsymbol{h}_y^l(x',y')) = \frac{\boldsymbol{h}_y^l(\ldots, x_k, y_k, z_k, x, y) \cdot \boldsymbol{h}_y^l(\ldots, x_k, y_k, z_k', x', y')}{\|\boldsymbol{h}_y^l(\ldots, x_k, y_k, z_k, x, y)\| \|\boldsymbol{h}_y^l(\ldots, x_k, y_k, z_k, x', y')\|} \tag{2}$$

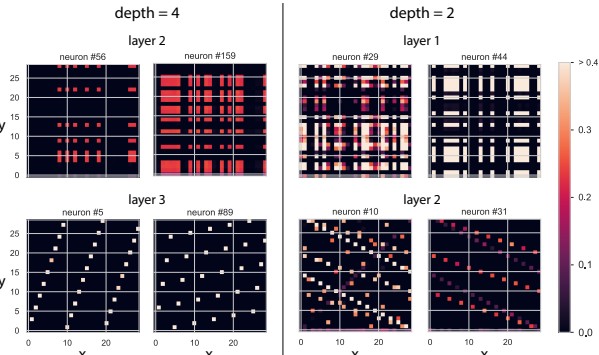

Figure 8: MLP activations (post-ReLU) wrt inputs $(x,y)$

We evaluate this similarity metric across all possible input pairs $(x,y)$ and $(x',y')$, resulting in a $p^2 \times p^2$ matrix. We also repeat the analysis to layer output vectors at $z$ tokens $\boldsymbol{h}_z^l(\cdot)$. In line with our previous analysis, we use controlled (fixed) in-context examples $(x_1\ y_1\ z_1\ \ldots\ x_k\ y_k\ z_k)$.

For the $d = 4$ model, the top panel of Figure 7 reveals a distinctive kaleidoscopic pattern[3] of high cosine-similarities. The pattern corresponds to input pairs $(x,y)$ that share the same ratio $y/x \bmod p$. In the MLP modules, we find highly structured activations as functions of inputs $(x,y)$ (Figure 8 left). Together with our earlier analysis of Skill I, these observations suggest that the MLP layer following the attention module with a "circle-of-circles" head (layer 2, head 4 for $d = 4$ model, see Figure 13 (a)) implements division operations over $\mathrm{GF}(p)$. The bottom panel of Figure 7 succinctly demonstrates the algorithmic shift from Ratio Matching to Modular Regression as the number of in-context examples increases. At 2-shot, layer 4 features collapse to identical vectors except when the ratio $y/x$ matches one of the ratios ($y_1/x_1$ or $y_2/x_2$). At higher shots, we see a transition to a task-dependent pattern where features align for input pairs $(x,y)$ for which $a\,x + b\,y \bmod p$ match – a signature of Modular Regression.

For $d = 2$ models, we again observe the kaleidoscopic pattern in cosine-similarities (Figure 7 top-right) and structured MLP activations (Figure 8 right). However, in this case we only find signatures of Ratio Matching in the cosine similarities (Figure 7 bottom-right), as expected.

Thus, we conclude that the $d = 2$ model has scarcely acquired skill III, due to its limited capacity. On the other hand, $d = 4$ model is very good at combining equations via skill III, explaining its superior performance. For a more detailed discussion, see Appendix E.

## 6 Discussion

We have investigated the emergence of in-context learning and skill composition in autoregressive models on a novel algorithmic dataset. The dataset includes a large discrete set of modular arithmetic tasks and is specifically designed to force models to learn how to solve a variety of tasks. It consists of learning linear modular functions, where the model is expected to identify *and* perform a modular operation in-context. When the number of tasks is relatively small the models can only generalize in-distribution. Although such models develop ICL capabilities, they simply memorize these task vectors and use them to classify the input vectors. Once the number of training tasks becomes too large, the models transition to a qualitatively different algorithmic approach, where the task vector is determined at inference time.

Finally, we have examined the learnt representations and shown that qualitatively different circuits are formed in different phases. In the o.o.d. generalization phase, we explain the learnt algorithms and highlight an in-context algorithmic shift in deeper models.

**Limitations**   We have limited this work to algorithmic datasets. It remains to be investigated what lessons can be translated to realistic language models, and what lessons are specific to the current setting. White-box interpretability analysis of deeper models has proved to be much more difficult than that of shallower models. Consequently, we still do not understand the role of every individual component of the network in the deeper cases.

---

[3] Patterns are cropped due to file size limit, check the GitHub repo for the full kaleidoscopic pattern.

## Acknowledgments

T.H. thanks Yue Xu and Dayal Singh Kalra for helpful discussions. A.G.'s work at the University of Maryland was supported in part by NSF CAREER Award DMR-2045181, Sloan Foundation. The authors acknowledge the University of Maryland supercomputing resources (`http://hpcc.umd.edu`) made available for conducting the research reported in this paper.

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

# A Experimental Details

## A.1 Model and Training Hyperparameters

**Architecture**  We used GPT-like architectures with Rotary Positional Embedding ($\theta = 10,000$) and ReLU activations. We fix the number of heads $H = 4$, embedding dimension $d_{\text{embed}} = 512$ and MLP widening factor 4 throughout every model. We use $d = 2, 4, 6$ throughout the paper. Embedding layers and the output layer are tied via weight tying.

**Initialization**  All linear layers and embedding layer weights are sampled from Gaussian distribution $\mathcal{N}(0, 0.02^2)$ at initialization, with the exception that the last linear layer in each MLP is sampled from $\mathcal{N}(0, 0.02^2/2d)$. No bias is used in any layer.

**Optimization and Schedule**  We trained most models using AdamW optimizer with learning rate $\eta = 1.5 \times 10^{-4}$, weight decay $\lambda = 2.0$, $\beta_1 = 0.9$, $\beta_2 = 0.98$, $\epsilon = 10^{-8}$, batch size $B = 1024$, $n_{\text{ct}} = 32$ in-context examples for 200k steps, together with a 5% linear warmup starting from $0.01\eta$ and a cosine annealing towards the end to $0.1\eta$. Weight decay is not applied to LayerNorm layers.

**Hyperparameter Choice**  For $d = 2$ models we scanned learning rates $\eta \in \{7.5 \times 10^{-5}, 1.5 \times 10^{-4}, 3 \times 10^{-4}, 6 \times 10^{-4}\}$ and weight decay values $\lambda \in \{0.5, 1.0, 2.0, 5.0\}$. Then we transfer our hyperparameters to other depths. Benefiting from the extra scaling in the initialization of the last linear in MLP, we find that the hyperparameters perform well for other depths. For larger $p$ values, we lowering down the learning rate to $10^{-4}$.

## A.2 Further Details of Each Plot in the Main Text

Figure 1 (b) Phase diagram constructed using $d = 6$ data shown in Figure 27, threshold for defining each phase is set to 75% for the corresponding set; (c) accuracy - number of shots curve for $d = 4$, $n_{\text{i.d.}} = 2^8$, $p = 29$ and $\alpha = 0.7$. (d, e) $d = 2$ model with $p = 29$, $n_{\text{i.d.}} = 2^9$ and $\alpha = 0.6$.

Figure 3 (a) Selected the best out of three random seeds with early stopping. (b, c) averaged over three random seeds with standard error labeled. All o.o.d. data are measured every $1,000$ step for 16 randomly sampled sequences along the pre-training. (d, e) Used checkpoint at the end of pre-training, averaged over 128 random sequences sampled from $S_{\text{test}}^{\text{o.o.d.}}$.

Figure 4 (a, b) Both phase diagrams are the best selected from three random seeds with early stopping. The loss - number of shots curves are plotted with $\alpha = 0.6$.

Figure 5 We use a $d = 2$ model trained with $\alpha = 0.6$ and $n_{\text{i.d.}} = 2^9$, and a $d = 4$ model trained with $\alpha = 0.8$ and $n_{\text{i.d.}} = 2^9$ with batch size $B = 1536$.

Figure 6 (a, b) Attention heads extracted form $d = 2$ model trained with $n_{\text{i.d.}} = 2^9$ and $\alpha = 0.6$. (c, d) Both models are trained with $\alpha = 0.6$.

# B Details of resources used

To generate each phase diagram and training curve, we used 90 GPU days on NVIDIA A100 40GB, with automatic mixed precision (BFloat16) and Flash Attention implemented in the PyTorch library[23]. During the exploring stage, we also used around another 90 GPU days. Most inferences were running on 1/7-NVIDIA A100 40GB GPU, of which the cost is negligible compared to the pre-training cost.

# C Table of log map for $p = 29$

| $n$ | 0 | 1 | 2 | 3 | 4 | 5 | 6 | 7 | 8 | 9 | 10 | 11 | 12 | 13 | 14 |
|---|---|---|---|---|---|---|---|---|---|---|---|---|---|---|---|
| $\log_{27}(n)$ | $0^4$ | 28 | 15 | 19 | 2 | 22 | 6 | 12 | 17 | 10 | 9 | 11 | 21 | 18 | 27 |

| $n$ | 15 | 16 | 17 | 18 | 19 | 20 | 21 | 22 | 23 | 24 | 25 | 26 | 27 | 28 |
|---|---|---|---|---|---|---|---|---|---|---|---|---|---|---|
| $\log_{27}(n)$ | 13 | 4 | 7 | 25 | 23 | 24 | 3 | 26 | 20 | 8 | 16 | 5 | 1 | 14 |

# D  Task Selection and Sequence Design

**Task Selection**  During the initial exploration phase, we observed that it was challenging for the model to learn multiple modular arithmetic tasks simultaneously. Typically, the loss would stagnate at a plateau indefinitely.

We hypothesize that when the model is trained on multiple modular arithmetic tasks, the strong bias inherent in each individual task may interfere significantly. When tasks are selected randomly, the resultant noise appears to inhibit the learning capabilities of the model, preventing it from acquiring any meaningful patterns or rules from the data.

Ultimately, we adopted a more structured approach to task selection as illustrated in Figure Figure 2, based on the following rationale: If the model is to learn a specific task vector $(a^t, b^t)$, it would presumably be more straightforward if one of the two components – either $a^t$ or $b^t$ – has already learned by the model. This would leave the model with only the other component to decipher, which we assume is a comparatively simpler task. Thus, we decided to employ the rectangular rule for sampling task vectors, which we believe strategically reduces the complexity of the learning process by partially leveraging previously acquired knowledge.

In Figure 9, we show an ablation plot of phase diagrams with a randomly selected task vector collection for pre-training. We still use the special sequence design.

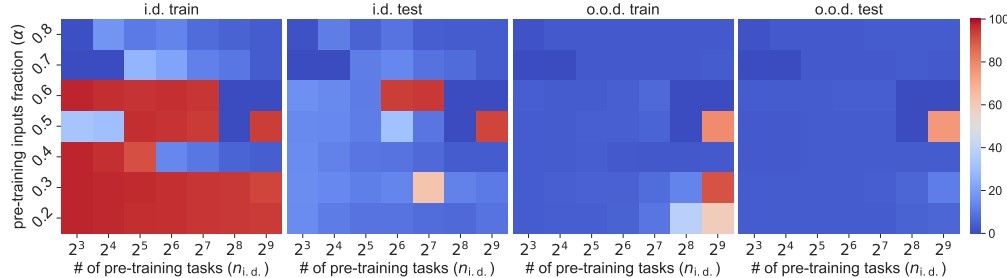

Figure 9: $d = 6$ phase diagram with task vectors randomly selected, selected the best results from two random seeds. Learning rate $\eta = 10^{-4}$.

**Sequence Design**  Following a similar spirit, we use a balanced batch where sequences generated from all task vectors appear exactly the same number of times during the training. We further align the examples across sequences generated by different task vectors, which we believe reduces the chance of the model getting confused by the same input $(x, y)$ appearing at different positions within the batch. Without this design, we could not make the model train.

Here, we also show the training curve per task in Figure 10, trained with all of our tricks. We see that the model first learned very few tasks and then eventually found its way out. For training without the task selection and sequence design, the loss typically plateau around $\sim 3.0$.

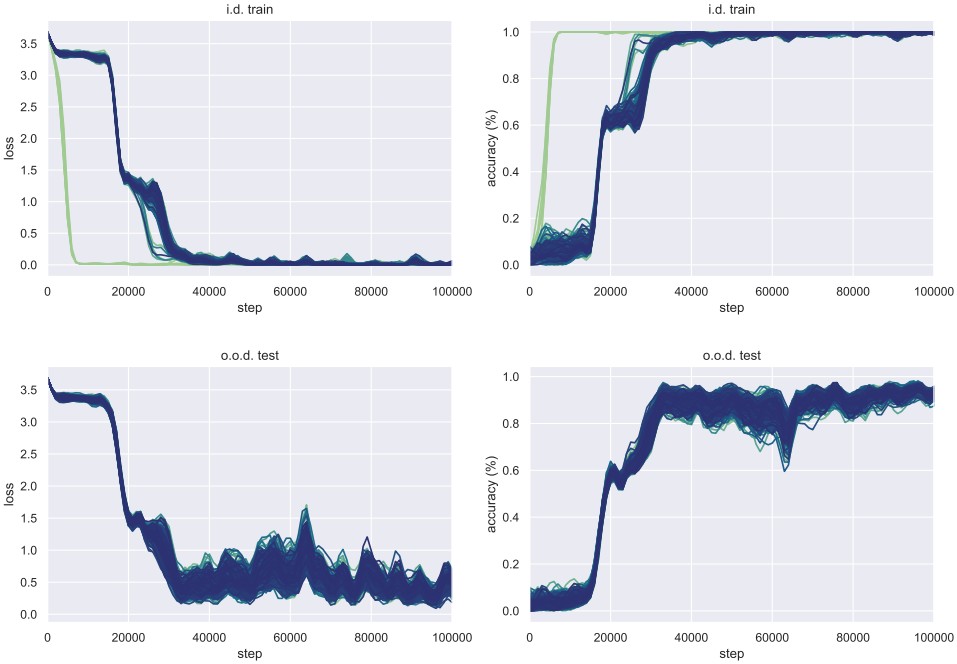

Figure 10: $d = 4$ training curve, with loss and accuracy measured per task for the last in context example. Trained with $n_{\text{i.d.}} = 2^8$ and $\alpha = 0.8$. There are six tasks learned almost instantly: $(5, 26)$, $(21, 23)$, $(28, 27)$, $(3, 9)$, $(10, 10)$, $(5, 9)$.

# E    Additional Interpretability Results

In this section, we show additional results on interpretability. Where we continue our discussion on how the algorithm might be implemented inside the model. Importantly, This includes PCA analysis of embedding (Appendix E.1), other attention heads (Appendix E.2), attentions and MLPs outputs (Appendix E.3), and finally, the role of MLPs (Appendix E.4). We think it is also crucial to study LayerNorm layers, which we leave for future work.

## E.1    PCA over Embeddings

We begin the discussion with the role of embedding layers. By further examining different models, we find highly structured embedding from $d = 2$ models. $d = 4$ models, on the other hand, does not have such structured embedding layers.

First we focus the embedding of $d = 2$ models. As shown in Figure 11, clearly, the logarithm of each number is split into even and odd groups, and each group forms one circle, which is a suitable embedding for doing modular multiplications. However, one should note that this will not obviate the importance of the head shown in Figure 6(a), as the model still needs a way to distinguish the same number that appears in the different positions in the context.

Curiously, we could not find such a structured embedding for the $d = 4$ model, as shown in Figure 12. However, as we will show in the next subsection, this non-structural embedding, together with the first layer, prepares a foundation for the essential heads in the latter layer to create a similar "circle-of-circles" feature as we shown in Figure 6 (a, b).

## E.2    Attention Heads

### E.2.1    $d = 4$ Model

To continue the story, we analyse the attention heads in different models. We first study the $d = 4$ model, where we also find a similar head with "circle-of-circles" (Figure 6(a)). Moreover, two other

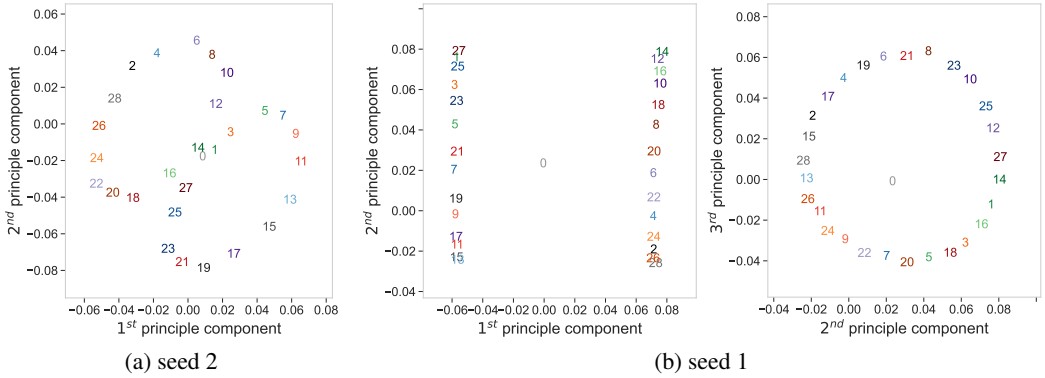

(a) seed 2    (b) seed 1

Figure 11: PCA over the embedding layers of $d = 2$ models with different random seeds. Each number is annotated by its $\log_{27}$ value.

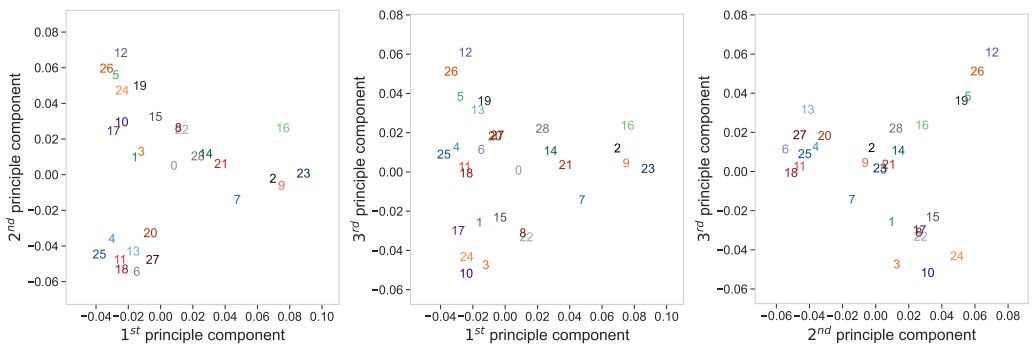

(a) PCA over the embedding layer for top-3 principal components

Figure 12: PCA over the embedding layers of $d = 4$. Each number is annotated by its $\log_{27}$ value.

heads put together are seemingly equivalent to Figure 6(b). We surmise that any model that solves modular arithmetic in-context requires such heads.

From Figure 13(a), we see that the head still pays attention locally within three token positions. Importantly, it also creates a circle-of-circles while performing PCA over concatenated $(x, y)$ features. The difference here is that the circle winds twice to go back to its origin. We believe that this factor of two differences in the period means that this head is effectively combing the row of the embedding layer of the $d = 2$ model and the role of the head in Figure 6(a). Overall, we do not yet know why the model needs to split the logarithm of numbers into even and odd groups.

From Figure 13(b, c), we find two heads that are structured but different from Figure 6(b). The PCA pattern forms circles while annotated within logarithm space, but depends on the choice of the sequences. This hints that those two heads are trying to exchange information with the previous inputs. We leave PCA analysis across different in-context examples to future work.

### E.2.2 $d = 2$ Model

Next we focus on the $d = 2$ model. In Figure 14, we plot similar PCA to the one in Figure 6(a), but with different concatenated features from the same heads: $(x, z)$ and $(y, z)$. We again see circle-of-circles, solidifying our argument that this head provides proper spaces for later heads to perform modular operations.

In Figure 15, we dump all the attention patterns we had for $d = 2$ models. The corresopnding PCA over concatenated $(x, y)$ features are shown in Figure 16 (except for layer 1 head 4, for which we plot for $(x, z)$). The special head we choose here has a similar behavior to the one in Figure 6(a), with its main focus shifted by one position but still within two preceding tokens.

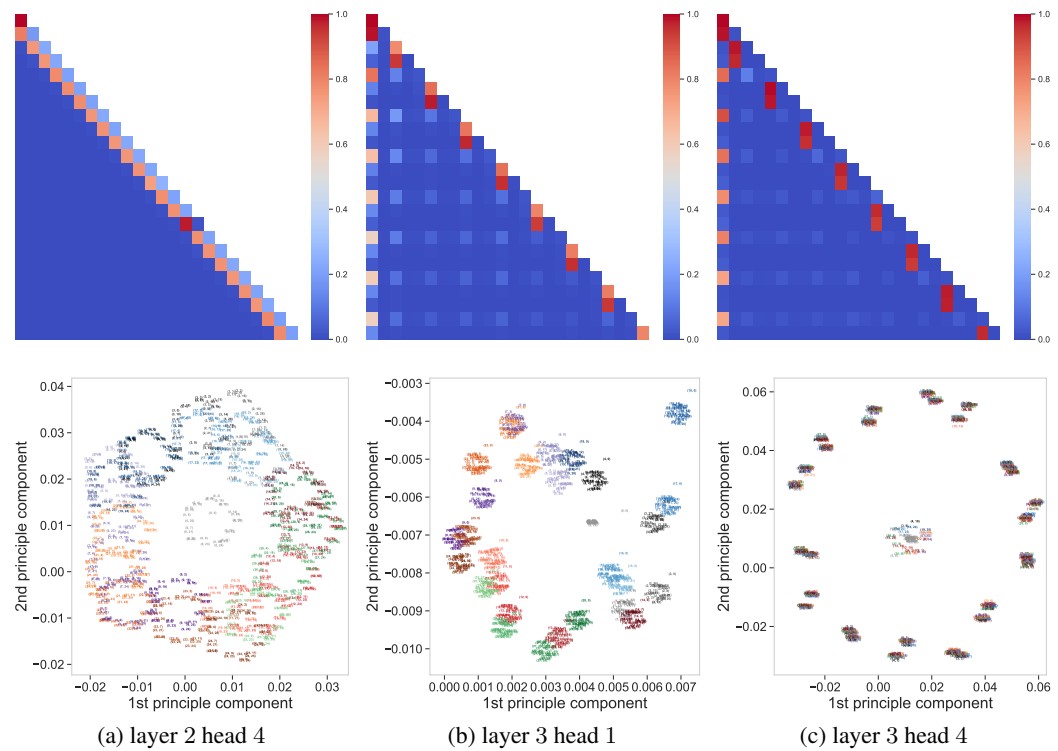

(a) layer 2 head 4       (b) layer 3 head 1       (c) layer 3 head 4

Figure 13: PCA over the features of specific heads for $d = 4$ model. Each number is annotated by its $\log_{27}$ value. We plotted PCA over (a, b) $(x, y)$ and (c) $(y, z)$ concatenated features. Where the circle in (b, c) heavily depends on the sequence and (a) remains unchanged.

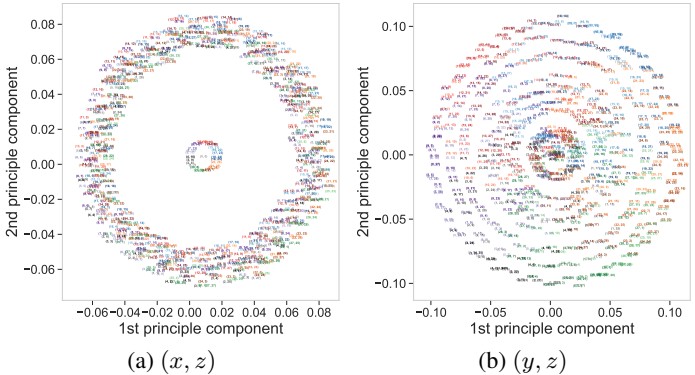

(a) $(x, z)$       (b) $(y, z)$

Figure 14: PCA over the same head in Figure 6(a), with feature concatenated differently. We annotate numbers with their corresponding logarithm.

In Figure 17, we extend our PCA analysis to different task vectors and up to top-4 components, where the spatial structure of the features is better depicted.

We have reasons to believe that the non-circle heads are not essential to the models' performance. However, we leave a careful exploration of this front for future work.

### E.3   PCA over Attention and MLP Outputs

In Figure 18, we plot $d = 2$ PCA analysis up to top-4 components for attention and MLP outputs.

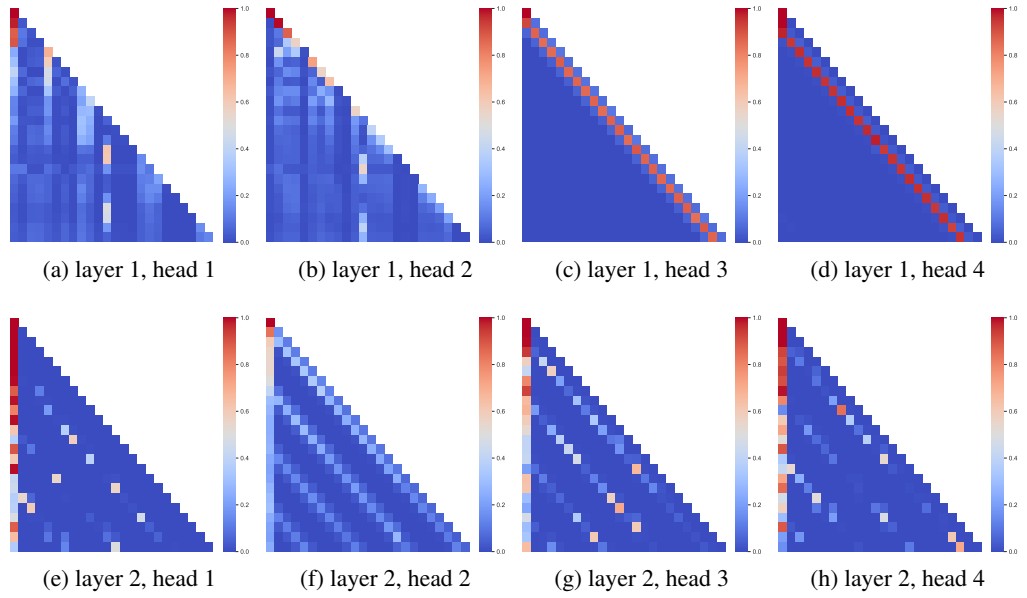

Figure 15: All attention heads in $d = 2$ model.

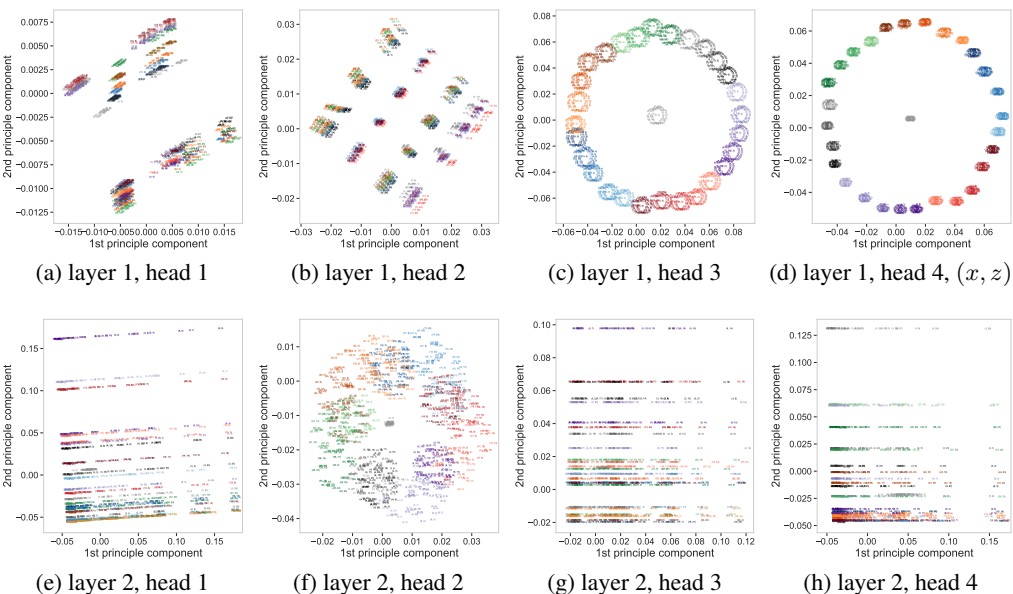

Figure 16: PCA for concatenated $(x, y)/(x, z)$ features for all heads in $d = 2$ model.

### E.4 Cosine-similarity analysis for MLPs

As we mentioned in Section 5.2, Skills II and III are likely implemented within the MLP layer. While we provided extensive details on how the signature of Skill II was observed in MLP, we could not find clear signals similar to those in Nanda et al. [20], Gromov [11] from MLP layers on Skill III.

#### E.4.1 $d = 4$ model

Here, we show an extended version of Figure 7 for all layers from the $d = 4$ model. We plot for 2-shot (Figure 20) and 16-shot (Figure 21), the task vector used is $(a, b) = (2, 6)$.

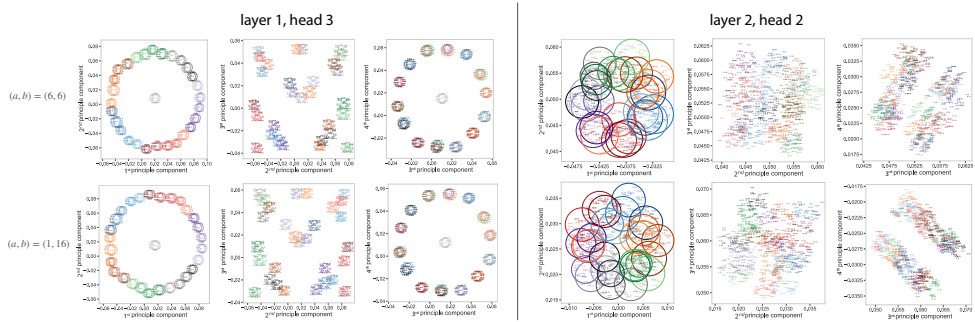

Figure 17: More PCA analysis similar to Figure 6. Top-2 components of PCA contribute $\sim 40\%$ variance of the layer 1 head and $\sim 20\%$ variance of the layer 2 head, while the top-4 components contribute $\sim 60\%$ and $\sim 40\%$ accordingly.

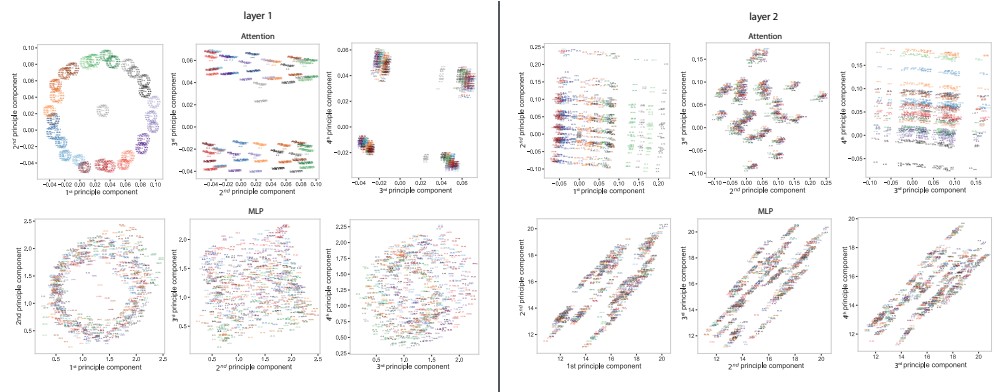

Figure 18: PCA analysis of Attention and MLP outputs. Attention modules contribute $\sim 50\%$ and $\sim 30\%$ to top-4 components of PCA of layer 1 and layer 2. MLP modules contribute $\sim 10\%$ and $\sim 12\%$ accordingly.

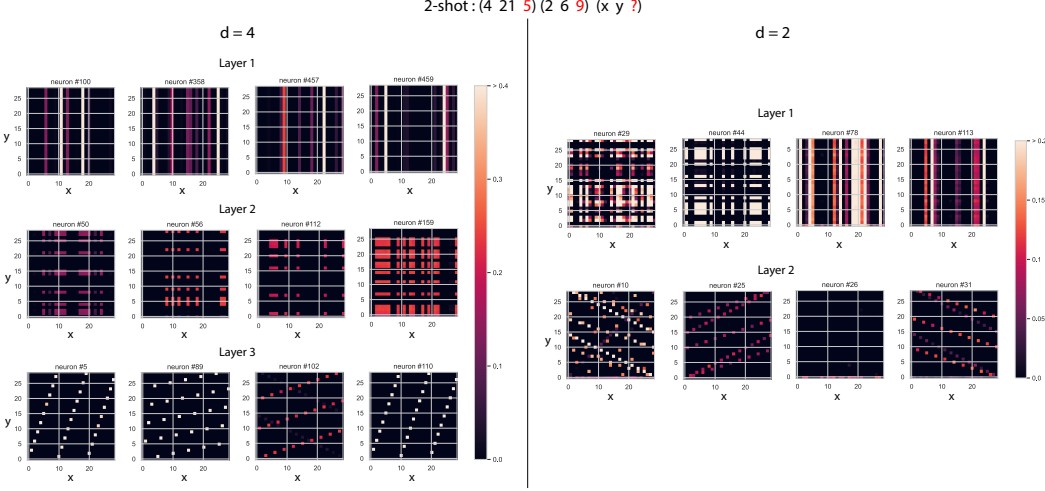

Figure 19: MLP (hidden) activations as a function of inputs $(x, y)$ for $d = 2, 4$ models.

### E.4.2 $d = 2$ model

Here, we show an extended version of Figure 7 for all layers from the $d = 2$ model. We plot for 2-shot (Figure 22) and 10-shot (Figure 23), the task vector used is $(a, b) = (2, 6)$.

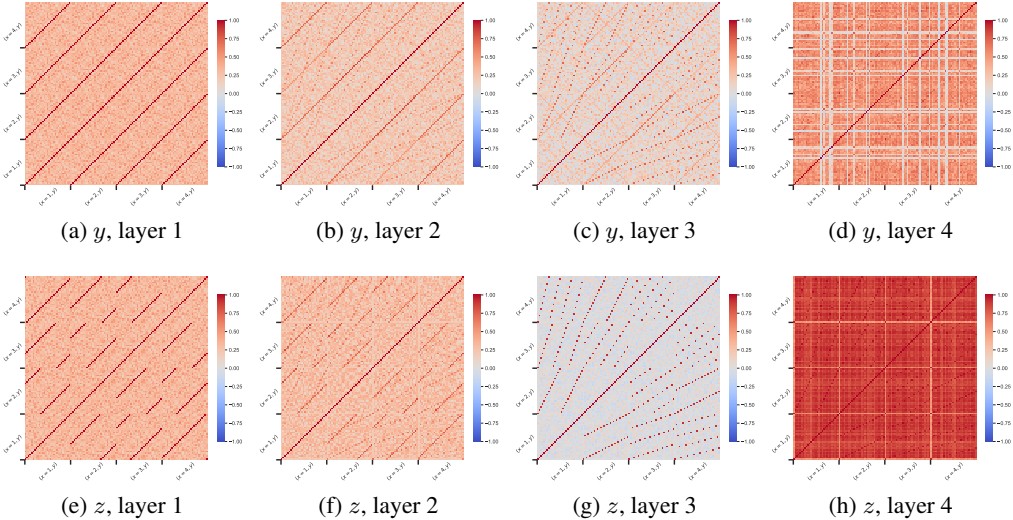

(a) $y$, layer 1     (b) $y$, layer 2     (c) $y$, layer 3     (d) $y$, layer 4

(e) $z$, layer 1     (f) $z$, layer 2     (g) $z$, layer 3     (h) $z$, layer 4

Figure 20: Cosine-similarity of $d = 4$ model for every block, measured at 2-shot.

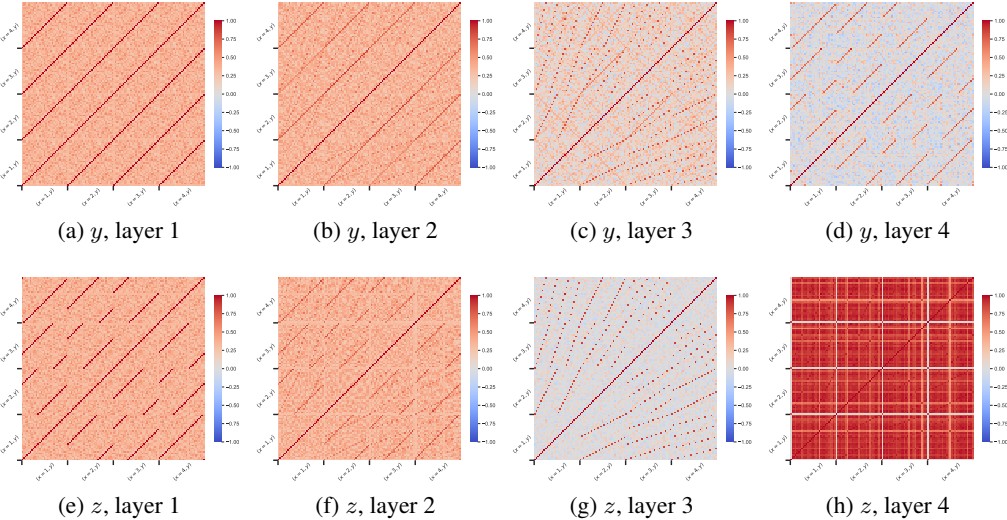

(a) $y$, layer 1     (b) $y$, layer 2     (c) $y$, layer 3     (d) $y$, layer 4

(e) $z$, layer 1     (f) $z$, layer 2     (g) $z$, layer 3     (h) $z$, layer 4

Figure 21: Cosine-similarity of $d = 4$ model for every block, measured at 16-shot.

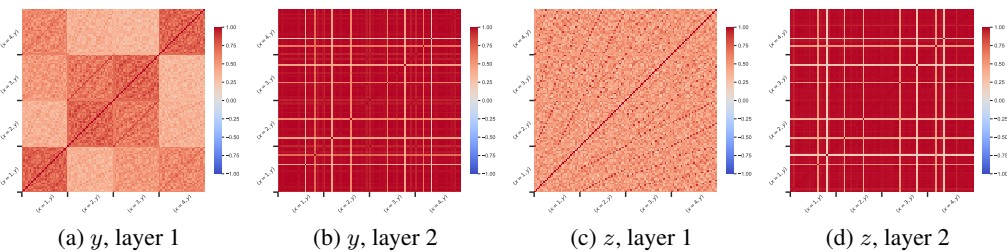

(a) $y$, layer 1     (b) $y$, layer 2     (c) $z$, layer 1     (d) $z$, layer 2

Figure 22: Cosine-similarity of $d = 2$ model for every block, measured at 2-shot.

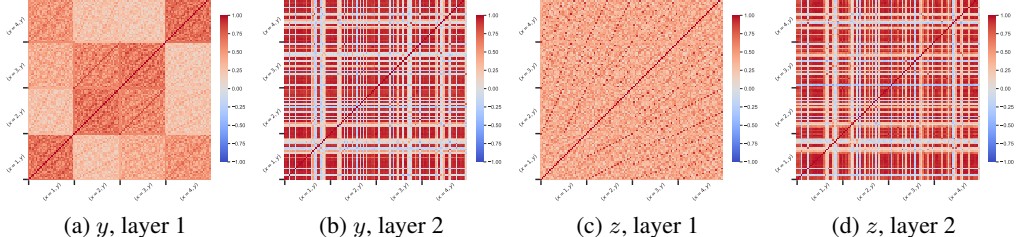

(a) $y$, layer 1    (b) $y$, layer 2    (c) $z$, layer 1    (d) $z$, layer 2

Figure 23: Cosine-similarity of $d = 2$ model for every block, measured at 10-shot.

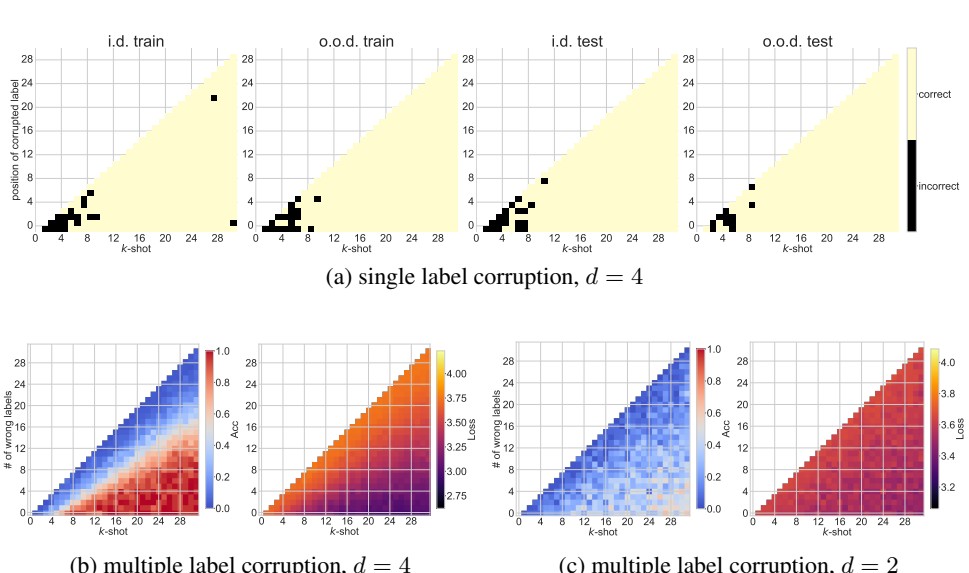

(a) single label corruption, $d = 4$

(b) multiple label corruption, $d = 4$    (c) multiple label corruption, $d = 2$

Figure 24: **(a)** Single label corruption : Performance of last token prediction as a function of changing position for a single label corruption. The x-axis shows different shots presented to the model, while the y-axis shows the position $j$ where the label is corrupted $z_j \to z'_j$. The $d = 4$ model is remarkably robust for long sequences, indicating the use of an algorithm which is not sensitive to the position of preceding examples. Qualitatively similar plots hold for other task vectors as well. **(b, c)** Multiple label corruption, at random locations for $d = 4$ and $d = 2$ models respectively. The $d = 4$ model is more resilient to label corruption than the $d = 2$ model, implying that the algorithm the latter employs is imperfect due to limited capacity.

## E.5 Label Noise

To gain insight into how the model combines the in-context examples, we introduce label-corruption in the in-context examples. In particular, we note the effect of (i) amount and (ii) position of label corruption on the model's performance. When we corrupt a *single* in-context example for $d = 4$ model, the model performance remains unaffected for longer sequences. This hints at that weighted average of the in-context inputs being used in model prediction. The $d = 2$ model, however, did not show such resilience.

Next, we corrupt *multiple* in-context examples in random locations. We study the effect on model performance as the amount of corrupted labels increases. While the $d = 2$ model is easily overwhelmed, the $d = 4$ model is able to offer strong resistance even at $\sim 40\%$ label corruption, for long sequences. This behavior remains invariant with the change in task vector for the particular sequence, indicating the universality of the underlying algorithm necessary for o.o.d. generalization.

# F    Additional Training Curves

We plot some selected training curves for $d = 4$ (Figure 25) and $d = 2$ (Figure 26) from Figure 4 phase diagrams. We see that even for $d = 4$, ICL can be a transient. With increased $\alpha$ or $n_{\text{i.d.}}$, the transient nature goes away.

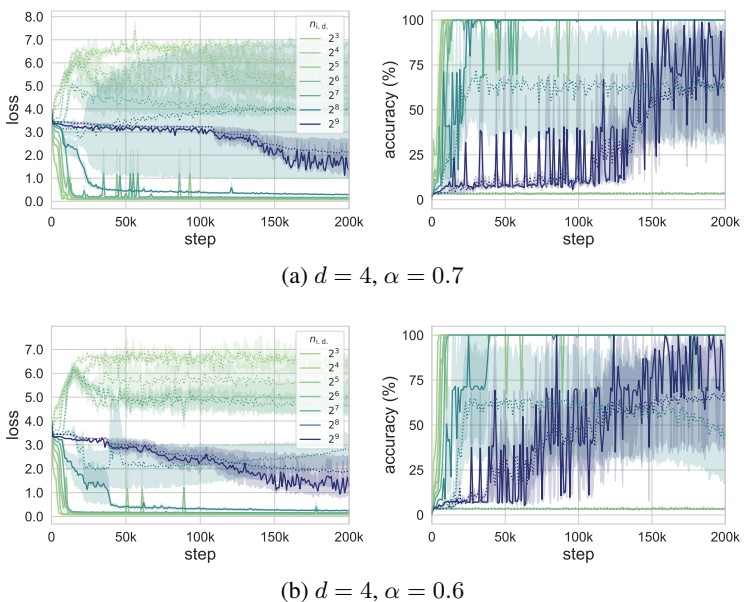

(a) $d = 4$, $\alpha = 0.7$

(b) $d = 4$, $\alpha = 0.6$

Figure 25: Training curves for $d = 4$, averaged over three random seeds.

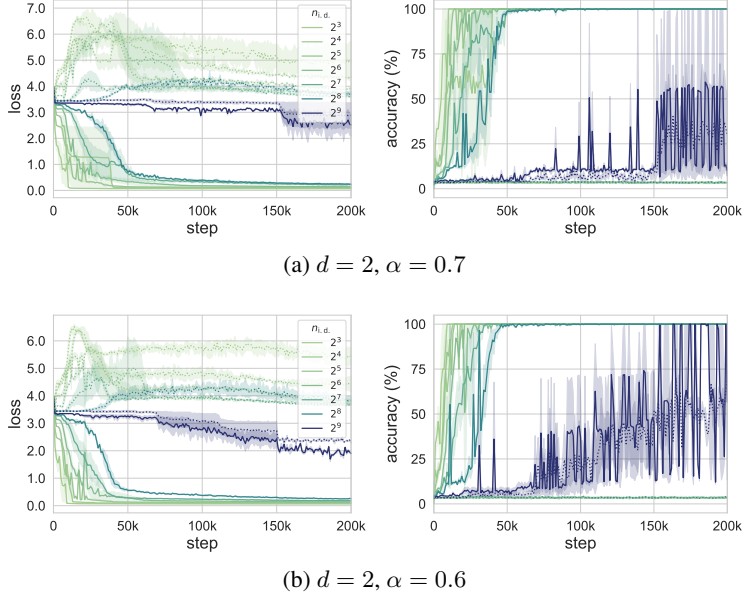

(a) $d = 2$, $\alpha = 0.7$

(b) $d = 2$, $\alpha = 0.6$

Figure 26: Training curves for $d = 2$, averaged over three random seeds.

# G Additional Phase Diagrams

In Figure 27, we plotted detailed/extended versions of the phase diagrams shown in Figure 1/Figure 4. The four phases story we have shown in Figure 1 still hold for other depths.

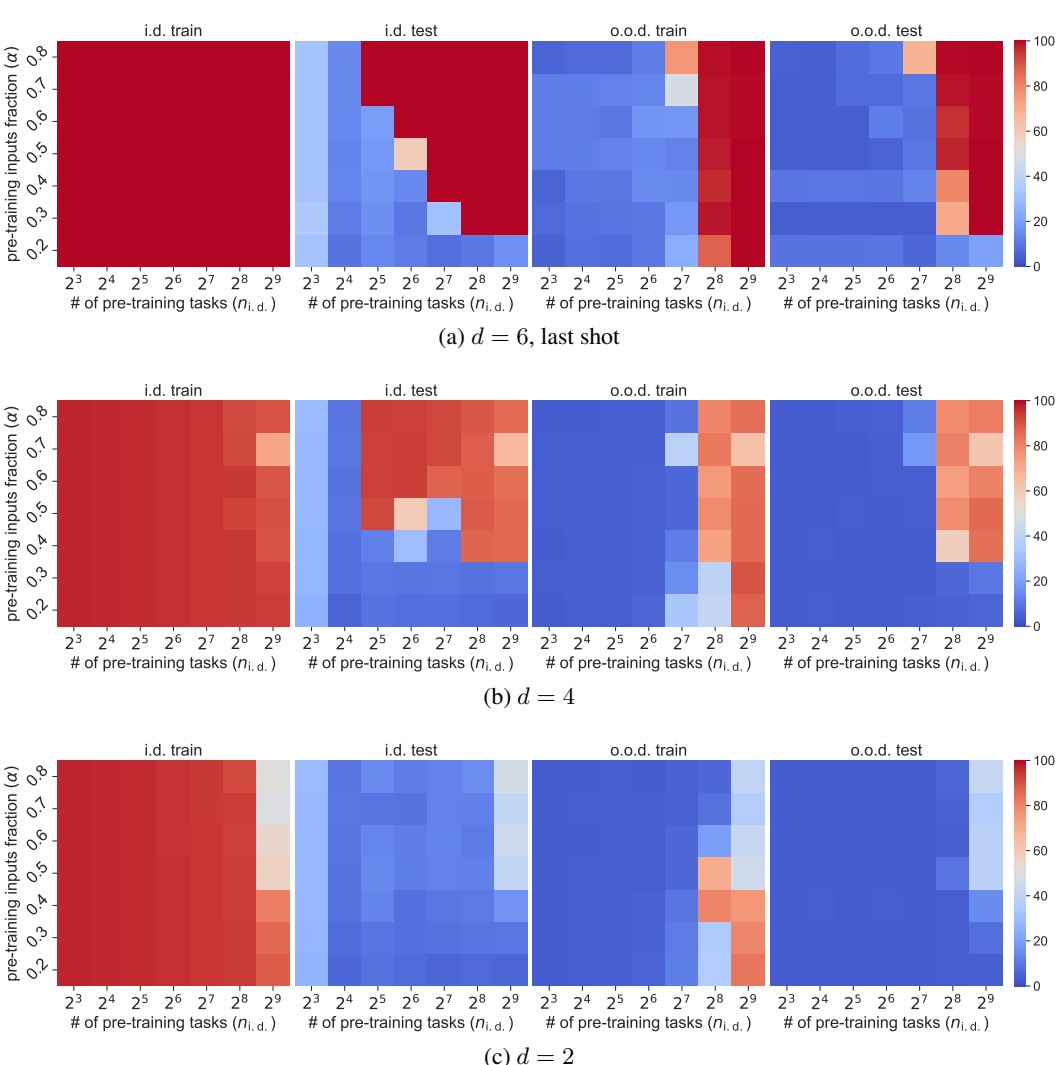

(a) $d = 6$, last shot

(b) $d = 4$

(c) $d = 2$

Figure 27: Phase diagrams on all four sets for $d = 4$ and $d = 2$.

# H Different Choice of $p$

In this section, we check the effect of varying task difficulties, i.e. the value of $p$. In Figure 28, we plotted o.o.d. generalization accuracy. Clearly as the task gets harder, the model needs to see more tasks to generalize out-of-distribution.

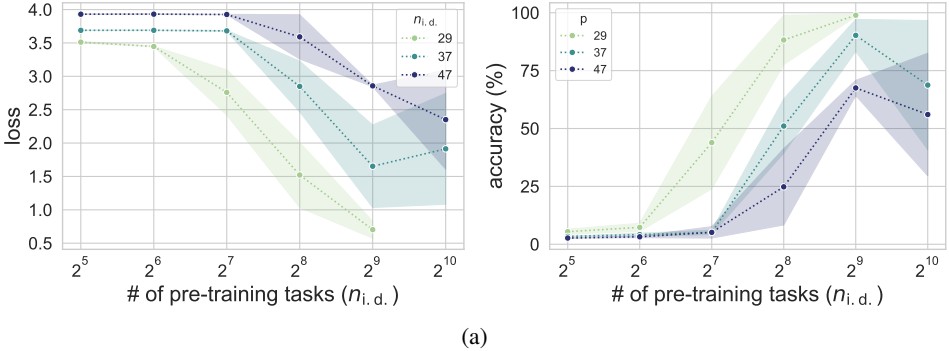

(a)

Figure 28: We compare the best performance for $d = 6$ models with different $n_{\text{i.d.}}$ values, averaged over three seeds. We use learning rate $\eta = 10^{-4}$ for $p = 37$ and $p = 47$, while keeping other hyperparameters the same as $p = 29$.

