# OpenReview forum: "Learning to grok: Emergence of in-context learning and skill composition in modular arithmetic tasks"
_NeurIPS.cc/2024/Conference — NeurIPS 2024 oral_

### Official Review · Reviewer_rwBm · 2024-06-14

**Soundness:** 3
**Presentation:** 4
**Contribution:** 3
**Rating:** 7
**Confidence:** 4

**Summary:**

The paper studies the emergence of the in-context ability of the GPT-style transformer model trained using autoregressive loss and arithmetic modular datasets. It analyzes the influence of the number of tasks, number of in-context examples, model capacity, etc., on the ICL capability of an appropriately trained model (i.e., using early stopping). It also provides a persuasive “task decomposition hypothesis”, which is well supported by the ablation study and various experiments. The white-box analysis on the attention heads provides convincing evidence of the proposed explanation. Although there is a gap between the grokking settings (i.e., small model and toy dataset) and practical systems, the paper does a good job of explaining many important trends and concepts related to the emergence of compositional in-context ability. I enjoy reading this paper and suggest an acceptance.

**Strengths:**

- The paper is easy to follow. Good presentation!
- The experiments are well-designed, providing compelling support for the claims.
    - The results in Figure 5 are cool.
    - The skill decomposition discussed in section 5 is great. The clear pattern in attention heads verifies it very well. (The hypotheses could be further verified if the author can link the values of $c_1, c_2$ to some weights in the network, see the question part.)

**Weaknesses:**

- The emergent ability (or grokking) usually refers to a phenomenon in the model “got stuck” in a non-generalization region and suddenly gained the generalization ability. Hence some discussion about the learning dynamics, i.e., how the accuracy, loss, representation, ability, attention pattern, etc., gradually evolve during training would make the paper stronger.
- The task and batch sample selection in this paper have many constraints (e.g., the rectangular rule, the balanced number of samples in each batch, etc.). However, the practical systems usually cannot strictly satisfy all these assumptions. Hence a more detailed analysis of how these assumptions influence the generalization ability would provide more insights to practical systems.

**Questions:**

- The paper claims in line 147 that “As the o.o.d. performance increases, the pre-training performance simultaneously degrades “. However, it is hard to read this information from Figure 3-a panel 1. Maybe a different color mapping or adding numbers on these patches would be helpful.
- Equation 2 is a bit hard to understand. How does it correlate to $z = ax+by$ ? (Although, from the latter explanations, I know the model relies on $c_1z_1^t + c_2z_2^t$ to get $z$, but it might be helpful to claim how it is derived.)
- Better to define $GF(p)$, i.e., the Galois field, before using it.
- Are the results in Figure 6 coming from $d=2$ or $d=4$? I can find the figure for all 8 attention heads for $d=2$ in the appendix, what about the $d=4$ case? It might be helpful to see if the pattern in later layers (i.e., attention focusing on different $z_i$) exists in shallow layers, and vice versa.
- In line 264, the paper claims that the pattern depends on $(a,b)$, but it is hard to read that from Figure 6b.
- As also mentioned in the strength part, is it possible to find some specific value in the weight space (e.g., attention weights, readout layers, etc.) that is highly correlated to $c_1, c_2$? If so, the hypothesis that the model first learns skill 2 (scale each example) and then skill 3 (weighted combine different examples) would be further verified.
- The OOD settings studied in grokking or emergent ability setting are quite related to the compositional generalization and systematic generalization. It would be helpful to discuss them in the related works, here are some of them:

    [1]  Schott, Lukas, et al. "Visual representation learning does not generalize strongly within the same domain." ICLR 2022

    [2] Xu, Zhenlin, Marc Niethammer, and Colin A. Raffel. "Compositional generalization in unsupervised compositional representation learning: A study on disentanglement and emergent language." NeurIPS 2022

    [3] Ren, Yi, et al. "Improving compositional generalization using iterated learning and simplicial embeddings." NeurIPS 2023

**Limitations:**

Discussions on how the findings help the practical system.

---

> ### Author Rebuttal · Authors · 2024-08-05
>
> We thank the reviewer for the encouraging feedback and incisive questions.
>
> ## Weaknesses
>
> **Emergent abilities / Grokking**:
> The loss and accuracy curves are already presented in Figure 3 of the current version of the paper. We agree that the gradual emergence of useful representations as a function of training time are useful results to showcase. In the final version of the paper, we will also include the feature analysis (similar to Figures 5,6) for intermediate checkpoints during training -- as suggested by the reviewer. (Note that we have not included these plots in the PDF of the Global Rebuttal due to space constraints.)
>
> We also note that we exercise a broader take on emergent ability than only learning dynamics. Emergent behaviours [1] are characterized by qualitative changes in model capabilities upon scaling-up (i) model size, (ii) dataset size, (ii) training duration, (iii) task-diversity etc. While the transition with training duration is an interesting aspect of the literature on Grokking, it is only one part of the emergent phenomena in deep learning. Moreover, careful initialization and optimization have been known to mitigate such effects [2]. However, the transition in capabilities with respect to dataset and model sizes are known to be more robust.
>
> **Pre-training task and batch selection**:
> The structured selection of tasks (rectangular rule) and balanced batches largely serve the purpose of making the pre-training more stable. Our intuition for using the specific setup was to reduce batch noise, as the training itself is challenging for this task. We strongly believe that scaling up the batch-size and model sizes will alleviate these constraints. We did not explore this avenue due to compute restrictions.
>
> ## Questions
>
> **o.o.d. vs pre-training performance**:
> The performance trade-off between o.o.d. and pre-training is more clear in the $d=2$ and $d=4$ models, shown in Figure 4. We will update line 147 and add a reference to the phase diagrams in Figure 4.
>
> **Equation 2**:
> Following the reviewer's suggestion, we will modify Equation (2) to read as follows:
> $$c_1 (x_1, \\; y_1 ) + c_2 (x_2, \\; y_2 ) = (x, \\; y) \\; \mathrm{mod} \\; p
> \qquad \xrightarrow{\text{find} \\; c_1, c_2} \qquad
> z = c_1 z_1^t + c_2 z_2^t \\; \mathrm{mod} \\; p \\; \qquad\qquad (2)$$
>
> For completion, we provide an explanation of the algorithm here: The model finds the right way to "linearly combine" $(x_1, y_1)$ and $(x_2, y_2)$ to equal $(x,y)$. The re-scaling factors $c_1, c_2$ in this linear combination can then be used to get the correct answer: $z = c_1 z_1^t + c_2 z_2^t$. Here is an intuitive way to think about the algorithm: Instead of directly solving for the unknowns $(a, b)$, the model treats the in-context examples $(x_1, y_1, z_1)$ and $(x_2, y_2, z_2)$ as vectors. The task then becomes to fill in the last component of a new vector $(x, y, ?)$. By aligning the first two components using coefficients $c_1$ and $c_2$: $c_1 (x_1, y_1) + c_2 (x_2, y_2) = (x, y)$ mod $p$, the model naturally finds out the correct way of aligning $z_1$ and $z_2$ with $?$, given the fact that they are constructed with a same underlying linear relation $a x + b y$ mod $p$.
>
> We will add a version of this explanation to the final version of the paper to enhance clarity.
>
> **Galois Field**:
> We thank the reviewer for pointing out the missing definition of Galois Field. We will include that in the final version.
>
> **Figure 6 and attention heads**:
> The results in Figure 6 are for $d=2$. We will specify that in the caption as well as the main text in the final version.
>
> In the Appendix of the final version of the paper, we will extend Figure 11 to include all the attention heads from the $d=4$ model. Unfortunately we cannot show them in the Global Rebuttal PDF due to space constraints.
>
> **Task dependence of Figure 6(b)**:
> In Figure G.2 of the attached PDF (Global Rebuttal) we present the PCA of attention heads for multiple tasks. Taking a close look at the first column in the layer 2, head 2 case, we see that the attention pattern is different for the two different tasks.
> % (Note that the layer 2, head 2 plots may seem denser than Figure 6(b). This is because we have included *all* the points here -- in Figure 6(b) we had only included points with even $x$ values, to keep the figure clean and interpretable.)
>
> **Re-scaling coefficients $\mathbf{c_1, c_2}$**:
> We tried using linear probing to extract information of $c_1, c_2$ from the residual stream, but the result is inconclusive. Please see the Global Rebuttal for more details.
>
> **Compositional and systematic generalization**:
> We thank the reviewer for pointing out the relevant references. We will include their relation to our work in the final version of the paper.
>
> [1] Wei et al.; "Emergent Abilities of Large Language Models"; arXiv:2206.07682 (2022)
>
> [2] Kumar et al.; "Grokking as the transition from lazy to rich training dynamics"; ICLR 2024

---

> > ### Comment · Reviewer_rwBm · 2024-08-08
> >
> > Thanks very much for the author's response. The new results are quite interesting. All of my concerns are well resolved. I confirm my evaluation and hope to see its new version.

---

### Official Review · Reviewer_PjEW · 2024-07-11

**Soundness:** 4
**Presentation:** 4
**Contribution:** 4
**Rating:** 8
**Confidence:** 4

**Summary:**

* The authors propose a synthetic sequence learning problem that I would call
  'in-context modular regression', an elegant generalisation of prior work
  studying modular addition and in-context linear regression.
* Using carefully constructed batches the authors are able to train
  transformer models to perform regression for a subset of tasks (weights)
  and a subset of inputs.
* The authors show that under some conditions on the data distribution and
  model architecture, the transformers not only achieve good performance on
  tasks and inputs included during training, but they also generalise to new
  tasks and/or new inputs. The authors document the conditions governing
  these generalisation capabilities in detail including showing phase plots
  and observing that in larger models, the generalisation properties are
  transient (they appear and then disappear across the training process).
* The authors postulate a breakdown of skills required to correctly perform
  the task. They effectively isolate and examine the abilities of their
  models to perform each component task. They also inspect the activations
  of each head and identify patterns suggestive of partial mechanisms
  underlying the generalising behaviour of the models.

**Strengths:**

I thank the authors for submitting their excellent work which stands to have a substantial impact in the science of deep learning.

* The work makes a meaningful contribution to an exceptionally important and
  interesting topic of the emergence of capabilities and internal mechanisms
  in deep learning.
* The setting and experiments neatly isolate and clearly demonstrate several
  interesting phenomena of emergence of capabilities and shifting in the
  solutions found by deep networks throughout training, contributing to the
  field's developing catalogue of examples of these phenomena.
* Moreover, the proposed synthetic problem is both rich and elegant. I expect
  this framework will become a fruitful test-best for follow-up work studying
  emergence phenomena, helping the field to improve our empirical and
  theoretical understanding of these phenomena.
* The authors also offer a partial behavioural and mechanistic analysis which
  is a solid starting point for a more detailed understanding of the learned
  structures that emerge in this setting.
* While some elements of the analysis are complex, the authors have done an
  exceptional job of clearly presenting their findings. I feel careful study
  of each section and figure in the main text was rewarded since there was no
  question that occurred to me that was not addressed in the authors' clear
  descriptions or figures.
* The authors have acknowledged all of the related work that I am aware of.

**Weaknesses:**

I have not noticed any weaknesses in the paper that would temper my overall
recommendation to accept. However, I note the following weaknesses, some of
which the authors have already acknowledged, and others which they may like
to take into consideration if they are interested to improve the paper
further.

1. **Delicate training set-up.** The authors explain that training
   transformers on multiple modular addition tasks crucially relies on
   following a delicately balanced batch construction methodology.
   I am left wondering if this batch construction methodology, as a further
   departure from the standard language modelling setting, has any other
   implications for the learning process that may affect the generality of
   the results.
   Note: This weakness is not decisive because the authors clearly document
   their training methodology and it's not *that* artificial anyway.

2. **The mechanistic analysis is only partial.** The authors admit that they
   have not been able to identify an end-to-end mechanistic model of how the
   trained transformers perform the task. This leaves their posited skill
   decomposition and partial mechanistic analysis open to the possibility
   that they are incomplete.
   Note: I think the contribution the authors have given in terms of the
   setting, the generalisation phenomena, and the partial skill decomposition
   and mechanistic analysis are already significant.

3. **Relationship to prior work.** The related work section does a good job
   of summarising the contributions of prior work in in-context linear
   regression and modular arithmetic in the context of transformer models.
   However, I feel that this section could be improved if the authors
   attempted to offer greater insight into the relationship between these
   prior works and the present work. For example, the authors have an
   opportunity here to informally describe the in-context linear regression
   and the modular addition problem settings that the newly proposed setting
   generalises.

4. I noticed some minor text errors as follows, which I expect the authors
   can easily correct.

    * Line 94: The notation $[1, p^2]$ to me suggests a closed continuous
      interval, whereas you appear to mean $\lbrace1, \ldots, p^2\rbrace$, also in some
      cases denoted $[p^2]$.
    * It seems that equation 2 should read $\ldots = (z_1^t, z_2^2) \mod p$
      and the equation on line 203 should read $c_1x + c_2y \mod p$. That is,
      $x$ and $y$ should swap places with $z_1^t$ and $z_2^t$. Is this indeed
      a mistake, or am I missing something?
    * In figure 6 (top row) there is a typo: "Qeury" on the vertical axis.
    * In line 445 there is a broken link.

I have not studied all appendices in detail.

**Questions:**

1. Why is the title 'learning to grok'?

    * Is this meant in the sense that the grokking of a modular addition task
      is occurring in-context? If so, this seems a little inaccurate, since
      the phenomenon analogous to 'grokking' seems to still be occurring
      during pre-training.
    * To be honest this part of the title has puzzled me since I first looked
      at the paper. Even if my understanding above is wrong and the title has
      an accurate interpretation, that I have failed to notice it might be
      one data point suggesting that if you are going for a title that is
      both short *and* informative, this might not be the right choice.

2. In the figure 1 caption, is it possible to offer a clearer summary of the
   difference between in-distribution generalisation and out-of-distribution
   memorisation? On my first read through, treating the figure and caption as
   an overview of the work's main results, I had trouble distinguishing these
   two concepts.

**Limitations:**

The authors transparently acknowledge all of the limitations I was able to
identify within the paper itself.

---

> ### Author Rebuttal · Authors · 2024-08-05
>
> We thank the reviewer for the encouraging feedback and valuable comments.
>
> ## Weaknesses
>
> 1. **Delicate training set-up**:
> The structured selection of tasks (rectangular rule) and balanced batches largely serve the purpose of making the pre-training more stable. Our intuition for using the specific setup was to reduce batch noise, as the training itself is challenging for this task. We strongly believe that scaling up the batch-size and model sizes will alleviate these constraints. We did not explore this avenue due to compute restrictions.
>
> We will add this clarification to the final version of the paper.
>
> 2. **The mechanistic analysis is only partial**: We have included additional results in the Global Rebuttal and the attached PDF, further strengthening our analysis. Notably, this includes highly structured neuronal activation patterns (Figure G.1). However, the end-to-end algorithm still remains an open question.
>
> 3. **Relationship to prior work**:
> We will incorporate the reviewer's suggestion into the camera-ready version of our paper.
>
> 4. **Minor Text Errors**:
> We have addressed all the minor errors pointed out by the reviewer, except one, which is not an error. Specifically, equation (2) and line 203 are not typographical errors.
> The algorithm we propose differs from the conventional method humans use to solve linear systems of equations, which involves explicitly computing the coefficients $(a,b)$ from the in-context examples. Instead of finding the unknowns $(a,b)$, the model finds the right way to "linearly combine" $(x_1, y_1)$ and $(x_2, y_2)$ to equal $(x,y)$. The re-scaling factors $c_1, c_2$ in this linear combination can then be used to get the correct answer: $z = c_1 z_1^t + c_2 z_2^t$. To emphasize this point, we have modified Equation (2), which now reads:
> $$c_1 (x_1, \\; y_1 ) + c_2 (x_2, \\; y_2 ) = (x, \\; y) \\; \mathrm{mod} \\; p
> \qquad \xrightarrow{\text{find} \\; c_1, c_2} \qquad
> z = c_1 z_1^t + c_2 z_2^t \\;\mathrm{mod} \\; p \\; \qquad\qquad (2)$$
>
>     Here is an intuitive way to think about the algorithm: Instead of directly solving for the unknowns $(a, b)$, the model treats the in-context examples $(x_1, y_1, z_1)$ and $(x_2, y_2, z_2)$ as vectors. The task then becomes to fill in the last component of a new vector $(x, y, ?)$. By aligning the first two components using coefficients $c_1$ and $c_2$: $c_1 (x_1, y_1) + c_2 (x_2, y_2) = (x, y)$ mod $p$, the model naturally finds out the correct way of aligning $z_1$ and $z_2$ with $?$, given the fact that they are constructed with a same underlying linear relation $a x + b y$ mod $p$.
>
>     We will add a version of this explanation to the final version of the paper to enhance clarity.
>
> ## Questions
>
> 1. On the origin of the title: The authors have always been of the opinion that the central theme in grokking is the formation of highly structured representations at the end of training. In fact, these representations can be viewed as a 1st order phase transition (in the proper statistical mechanics sense as explained in https://arxiv.org/abs/2310.03789).
> If we take this perspective then in the present work grokking happens in _two_ qualitatively different ways: (i) _as optimization time passes_ the model learns to solve the task in-distribution (and sometimes o.o.d.), which requires highly structured representations, and (ii) _as the number of in-context examples increases during inference_ the model performance steadily improves; completely solving the arithmetic problem with enough in-context examples. It is crucial that the predictions are conditioned on the sequence of in-context examples and this conditioning is _emergent_. A more accurate name could be "grokking grokking", but we decided to opt for a milder version.
>
> 2. We thank the reviewer for pointing out the cumbersome phrasing. We have edited the part of the caption in Figure 1 that distinguishes various phases to make it clearer. The part of the caption now reads:
>
>     > **(b)** Phase diagram for a six-layer model. We find four different phases. (1) in-distribution memorization: The model *only* performs well on tasks $(a,b)$ *and* examples $(x,y)$ from the training set -- it does not generalize on unseen examples or tasks. (2) in-distribution generalization: model generalizes on unseen examples $(x,y)$ but not on unseen tasks $(a,b)$. (3) out-of-distribution memorization: model generalizes on unseen tasks $(a,b)$ but only for examples $(x,y)$ it has seen during training. (4) out-of-distribution generalization: model generalizes on unseen tasks $(a,b)$ for seen as well as unseen examples $(x,y)$. We focus on investigating phase (4) in more detail.
>
>     Additionally, we will add a table clarifying the performance on the sets $S_{train}^{i.d.}, S_{test}^{i.d.}, S_{train}^{o.o.d.}, S_{test}^{o.o.d.}$ in the four different phases, on page 4 of the main text. We hope that this will help avoiding any possible confusion in the definition of the phases.

---

> > ### Comment · Reviewer_PjEW · 2024-08-08
> > **Thanks for clarifying and for your proposed improvements to an already strong paper**
> >
> > Thank you for clarifying especially my confusion around the proposed vector scaling approach to solving the task. The proposed revisions and the additional experiments will further improve an already strong paper. I maintain my confident recommendation that this paper should be accepted.

---

### Official Review · Reviewer_Jerg · 2024-07-13

**Soundness:** 3
**Presentation:** 3
**Contribution:** 3
**Rating:** 7
**Confidence:** 3

**Summary:**

This paper studies the emergence of in context learning and skill composition in autoregressive models. They create an algorithmic dataset to probe how autoregressive models use tasks learned during training to solve new tasks. They find that more training tasks lead to a generalizing / algorithmic approach instead of memorization.

**Strengths:**

- This work introduces a new algorithmic dataset (with modular arithmetic tasks) that force models to learn a variety of tasks. The work finds that when the number of tasks goes from small to large, the model transitions from memorization to generalization.
- This work has many interesting experiments. I found Section 5.2 (Attention Heads Implement Essential Skills) pretty interesting.

**Weaknesses:**

- The definition of task diversity is not well defined. Is the number of pretraining tasks truly indicative of task diversity? I think the paper could benefit from some justification of this assumption.
- The paper claims that for larger models, early stopping is necessary (line 52). While I appreciate that the authors used GPT-like architectures to reflect realistic settings, the architectures in the experiments are not that large. Even amongst popular open source models, the smallest are usually around 7B parameters.
- Many works in the continual learning and meta learning literature suggest that training on multiple tasks at once leads to better generalization. Perhaps it is worth including brief discussion on the connections between this point and the model’s ability to generalize ood which is predominantly determined by the number of pre-training tasks.

**Questions:**

Since multiplication can be viewed as repeated addition, isn’t skill 2 an extension of skill 3 (or can even be viewed as skill 3 composed with itself multiple times)? Is hierarchy of skills important here?

**Limitations:**

As acknowledged by the authors, this work is limited to particular algorithmic datasets.

---

> ### Author Rebuttal · Authors · 2024-08-06
>
> We thank the reviewer for their insightful comments.
>
> ## Weaknesses
>
> **Task Diversity**:
> Our definition of task diversity follows the existing works on in-context learning with linear regression, with a key difference:
> since our tasks are defined over a finite field, the total number of possible tasks (labeled by the task vectors $(a, b)$) is _finite_ and equals $p^2$. This differs from commonly discussed cases of linear regression, where the set of tasks is infinite (and, in fact, _continuous_). Consequently, the number of pre-training tasks (as a fraction of the total tasks $p^2$) is a natural measure of task diversity. There is a subtlety in that tasks may not be completely independent from each other, and a true definition of task diversity should include a reference to independence. However, it is not clear to what extent the model leverages this possible redundancy -- and similar point is also omitted in the works on linear regression. We decided to not go down that rabbit hole and defined task diversity in a naive way.
>
> That being said, we acknowledge that different ways of sampling the task sequences could also influence o.o.d. generalization. To address this, one might construct a phase diagram with an additional axis representing task sampling. However, this would require an order of magnitude more computations and a detailed multi-page discussion, making it impractical for our current study.
>
>
> **Early Stopping and Larger Models**:
> We appreciate the reviewer raising this point.
> First, we would like to clarify that the "larger model" mentioned in line 52 refers to the comparison between the $d=6$ model and the $d=2,4$ models used throughout the paper.
> Notably, these settings are sufficient to demonstrate our point, as the model's scale should be measured relative to the dataset size. The SoTA LLMs are pretrained on corpora much larger and diverse than the arithmetic tasks that we study in this work. We agree that larger-scale experiments would be necessary to transfer the insights gained from our study to modern LLMs. However, such experiments are far beyond our current capabilities due to limited GPU resources.
> Finally, the purpose of including details such as early stopping is to aid reproducibility of our results -- we do hope that the community will explore and generalize our setting.
>
> **Relation to meta-learning and continual learning**:
> We thank the reviewer for pointing out this interesting connection. It is indeed possible that some of the insights from our work finds connections to these areas. In the current version, we have cited one work [1] related to meta-learning. In the camera-ready version, we will include a more elaborate discussion with more references.
> We welcome suggestions for specific resources about continue learning that the reviewer has in mind relevant to our study.
>
> ## Question
>
> The reviewer raises an interesting point about the hierarchy of skills, and is correct in pointing out that multiplication can indeed be constructed from repeated additions.
> However, it is important to think from the model's perspective. We believe that *efficient* implementation of finite field operations by the model requires separate components to perform addition and multiplication. One intuitive way to think about this is that the models do not have sufficient depth to perform arbitrary repeated additions to construct multiplication. Instead, the models build correct representations to implement multiplication.
> Consequently, it is better to think of the numbers on a finite field $\mathrm{GF}(p)$, with distinct operations of addition and multiplication.
>
> [1] Louis Kirsch, James Harrison, Jascha Sohl-Dickstein, and Luke Metz; "General-purpose in-context learning by meta-learning transformers"; https://arxiv.org/abs/2212.04458 (2022)

---

> > ### Comment · Reviewer_Jerg · 2024-08-12
> >
> > Thank you for the detailed response. I have increased my score 6 --> 7.

---

### Official Review · Reviewer_CrUb · 2024-07-15

**Soundness:** 3
**Presentation:** 3
**Contribution:** 3
**Rating:** 7
**Confidence:** 4

**Summary:**

This paper develops novel insights into in-context learning and how it works in Transformers. To this end, the authors propose a generalization of the modular arithmetic task explored in several prior works on grokking. Unlike those works, the structure of the defined task is more rich, enabling an analysis of both in-distribution generalization (standard test evaluation) and out-of-distribution generalization (which is itself broken down into two variants).

**Strengths:**

The paper is fairly well written and clear. Going beyond the standard linear regression task to study ICL was great to see as well.

The main selling point for me are the empirics though---I really like the results! The visualization of how the model represents concepts relevant to this paper's setup is quite beautiful: the circle of circles was fascinating to look at and, arguably, not something I expected. In retrospect, I can rationalize this as making sense---we get circular embeddings in grokking, so circle of circles is the logical geometrical extension here. Results on scaling are interesting in their own right as well.

**Weaknesses:**

I do not have any major apprehensions, except for the related work, which I think is relatively sparse.

- **Related Work.** At this point, the topic this paper is focused on has a rather rich literature and I think a more detailed related work is warranted (perhaps in the appendix if space is an issue). For example, the results by Kirsch et al. (which is cited) are very similar to what authors show, especially results on scaling effects. The main different is width scaling in that paper and no geometric analysis, but nonetheless the relationship warranted more emphasis and discussion. Similarly, several recent works have explored OOD generalization of toy ICL tasks defined in prior works (e.g., see Ahuja and Lopez-Paz [1] for work on linear regression tasks and Ramesh et al. [2] for group arithmetic tasks). Regarding grokking, there are several works exploring the phase transition-y nature of this task. For example, see Kumar et al. [3]. The transient nature of ICL also has negative results (see Reddy [4]), which are worth discussion since they are the primary conclusion in depth scaling as I see it.

[1] https://arxiv.org/abs/2305.16704
[2] https://arxiv.org/abs/2311.12997
[3] https://arxiv.org/abs/2310.06110
[4] https://openreview.net/forum?id=aN4Jf6Cx69

**Questions:**

A few questions below that I would like to see answered.

- **PCA variance.** Given this is a rather rich geometry in 2-D, I'm slightly surprised to see PCA captured it. Did you have to do some preprocessing? How much variance is explained by the two projected components? If there are other components that are not shown but have a large variance, what do those components encode---can you try 3D plots?

- **What does the MLP do?** Given the mechinterp focused on attention solely, it is unclear what role MLPs played. Two experiments to try here are: (i) train attention only models to see if MLPs are even necessary, and (ii) perform the PCA analysis to uncover representations' geometry at the level of attentions and MLPs at each block in the model. Experiment (i) may require retraining models, so I understand if the authors are unable to conduct it, but my expectation will be that you will see that model "internalizes" task vectors and records them in MLPs. Attention only models can solve the task, but I expect the representations' geometry will be quite different. For experiment (ii) however, I expect that's easy to run and is merely repeating the plotting script on intermediate representations as a forward pass occurs through the model. If the geometry is primarily formed at attention layers, we'll see that in this experiment; vice versa, if it forms via MLPs, we'll see it explicitly.

**Limitations:**

Limitations are fairly discussed.

---

> ### Author Rebuttal · Authors · 2024-08-05
>
> We thank the reviewer for the encouraging feedback and helpful suggestions.
>
> ## Weakness
> We thank the reviewer for pointing out the highly relevant references. We will add the citations and utilize the additional page allowance in the final version to discuss their relation to our work.
>
> ## Questions
>
> **PCA Analysis**: We conducted PCA without any preprocessing. As per the reviewer's suggestion, we expanded upon Figure 6 from the original manuscript by plotting higher order PCA components. The results are presented in Figure G.2 of the attached PDF and discussed in the Global Rebuttal. We analysed the top-4 components, and found highly structured features. The top-4 PCA components account for a significant portion of the PCA variance. (Note that we present 2d slices instead of 3d plots of PCA because sparse 3d plots shown in 2d are difficult to comprehend.)
>
> Furthermore, following the suggestion of the reviewer rwBm, we plot similar features with a different task vector $(a,b)$ and a different number of shots. These results serve as evidence for the claims made in Figure 6 caption. Specifically, the PCA constructed from the first layer's head remains unchanged across different task vectors (up to a negative sign along certain directions). In contrast, the PCA derived from the second layer's head changes with the choice of task vector.
>
> **PCA of Attention outputs**: As per the suggestion of the reviewer, we present the PCA analysis of Attention outputs (as opposed to individual heads) in the top row of Figure G.3 of the attached PDF. We find highly structured top-4 PCA patterns in Layer 1, which also account for a significant fraction of PCA variance. Layer 2 exhibits less structured organization and the contribution of the top-4 PCA components is diminished.
>
> **Analysis of MLP features**: In the attached PDF, we have added two main results concering MLPs.
>
> 1.  We extended our analysis to include PCA Multi-Layer Perceptron (MLP) features, as shown in the bottom row of Figure G.3 of the attached PDF. The results demonstrate:
>
> - Layer 1: Highly structured patterns are evident in MLP features. The top-4 components contribute substantially to the overall PCA variance.
>
> - Layer 2: Features exhibit less structured organization, and the significance of the top-4 components is diminished compared to Layer 1.
>
> 2. Additionally, in Figure G.1 of the attached PDF, we have shown the post-ReLU neuronal activations from various layers as functions of $x,y$. We find highly structured activation patterns across layers, especially in Layer 3 of $d=4$ model. For a detailed account of the MLP results, please refer to the Global Rebuttal.
>
> We will discuss these new results in the additional page allowance of the final version. We believe that in a future work, our analysis of the attention heads as well as MLP activations can be tied together to infer an end-to-end algorithm for our setting.

---

> > ### Comment · Reviewer_CrUb · 2024-08-07
> >
> > Thank you for the response. I really like the new results and hope they'll be included in the final paper. I'll maintain my original score.

---

### Author Rebuttal · Authors · 2024-08-05

# Global Rebuttal

We included three new figures in the attached one-page PDF. These new results address questions raised by one or more of the reviewers. Especially, the results about MLP layers are relevant to multiple reviews.

## Figure G.1
We examined how individual neurons (post-ReLU) are activated for different inputs $(x,y)$. We discovered that each neuron only gets activated for highly specific inputs. This can be interpreted as a further skill composition at the neuron level, although the exact role of each neuron remains to be discovered.

Notably, in Layer 3 of the $d=4$ model we find that neuronal activations follow the re-scaling relation $x = k y$. The layer contains all such re-scalings, forming a complete basis. Layer 2 of the $d=4$ model show a periodic pattern wrt $(x,y)$, while Layer 1 neurons only get activated only for specific $x$ values.

Neurons in the $d=2$ model appear to be superposed/compressed versions of those found in the $d=4$ model. This is likely due to $d=2$ model not having enough layers. We observe that the neurons from Layer 1 of the $d=2$ model contains patters similar to Layer 1 and 2 of the $d=4$ model. Neurons from Layer 2 of the $d=2$ model appear to be superpositions of various re-scaling patterns from Layer 3 of $d=4$ model.

## Figure G.2
We expanded upon the PCA plots of the $d=2$ model presented in Figure 6 of the original manuscript. In this extension, we included a different task vector $(a, b)$ and plotted the results using a different shot (16-shot).

We see that The top-3,4 components of PCA of layer 1, head 3 also forms circle of circles, albeit with a different pattern from that of top-1,2 components. In this case, we find pairs of coinciding circles, where the $x$ corresponding to the coinciding circles differ by $(p-1)/2 = 14$.

In layer 2, head 2 we can see that the PCA pattern changes for different tasks. This is in contrast to the layer 1 PCA patterns, which remain unchanged. (as claimed in the main text)

## Figure G.3
We performed PCA on both (i) Attention output (as opposed to individual heads) and (ii) MLP output of the $d=2$ model. In the Attention output of Layer 1, we observe a circle-of-circle structure. The other components also exhibit some structure -- notably, the top-3,4 components in layer 1 form 4 clusters corresponding to even/odd $X$ and $y$ values.

## Additional Experiments

In addition to these results, we also ran a few more experiments. Due to space constraints, we could not include them in the one-page PDF. We describe these experiments and results in words here -- we will include them in the camera-ready version of the paper.

1. Linear probing of $c_1$ and $c_2$: We extracted the feature from the residual stream after each transformer block; attached a new linear layer and fine-tuned it to predict the correct re-scaling coefficients.

    We simplified the experiment to the $1$-shot case, where the sequences are simply $(x_1, y_1, z_1, x, y, ?)$. The fine-tuned linear layer was used to predict the correct coefficients $c_1$ such that $x_1 \cdot c_1 = x$ mod $p$ (alternatively $y_1 \cdot c_1 = y$ can be used). We observed $15 \\% - 20 \\%$ accuracy across all the layers for both $d=2$ and $d=4$ models. Despite the accuracy being above random guessing ($3\%$), we believe this result to be inconclusive.

2. Linear probing of (a,b): We also ran similar experiments with full sequences ($1$-shot to $31$-shots) and tried to predict the task vector $(a, b)$ along the sequence. We found random performance on the o.o.d. generalization set, suggesting that the model does not explicitly compute the task vector. Note that this is in agreement with our proposed algorithm.

3. Pruning / Activation Patching [1]: In this experiment, we replaced the output of each attention head with its averaged output over pre-training sequences. The average was taken over all the pre-training tasks, and $512$ sequences from each task. We found that:

    - For both $d=2$ and $ d=4$ models, pruning the circle-of-circle head immediately brings the model to random guessing. This can be understood as an average over sequences collapsing the circle down to a point, which destroys the feature completely.

    - For the $d=2$ model, pruning any other head causes some performance drop. This is to be expected since the model does not have enough capacity even before patching.

    - The $d=4$ model is significantly more robust to pruning. We can patch all heads except for (i) the three shown in Figure 11 of the manuscript and (ii) the heads in the last layer, with almost no impact on the performance (less than a $5\\%$ performance drop).

[1] Fred Zhang, Neel Nanda; "Towards Best Practices of Activation Patching in Language Models: Metrics and Methods"; https://arxiv.org/abs/2309.16042

---

### Decision · Program_Chairs · 2024-09-25

**Decision:**

Accept (oral)

**Comment:**

This paper studies the emergence of in-context learning and skill composition in Transformers. All reviewers were enthusiastic about the work and highlighted its novelty and generalization of prior work studying modular addition and in-context linear regression. The paper was praised for its impact on deep learning and its meaningful contribution to understanding emerging capabilities. Reviewers found the proposed synthetic problem elegant and a candidate for test-best for follow-up work. The experiments' design was appreciated because they underpinned the paper's claims and contributed to cataloging the emergent phenomena. Finally, the clarity of exposition and visualization was acknowledged, particularly due to some complex analysis elements. This translated to the reviewers' scores all exceeding 7 and a unanimous recommendation of acceptance.

This is by far the highest-rated paper from my batch (12 items), and due to such enthusiastic reviews, I recommend it to be considered for oral.

EDIT after SAC feedback: I highlighted the things that the reviewers praised.